# Global terrestrial moisture recycling in Shared Socioeconomic Pathways

Arie Staal[1], Pim Meijer[1,2], Maganizo Kruger Nyasulu[3,4,5], Obbe A. Tuinenburg[1], Stefan C. Dekker[1]

[1]Copernicus Institute of Sustainable Development, Utrecht University, Utrecht, 3584 CB, the Netherlands
[2]National Institute for Public Health and the Environment, Bilthoven, 3721 MA, the Netherlands
[3]Stockholm Resilience Centre, Stockholm University, Stockholm, SE-106 91, Sweden
[4]Potsdam Institute for Climate Impact Research (PIK), Member of the Leibniz Association, Potsdam, 14473, Germany
[5]Bolin Centre for Climate Research, Stockholm University, Stockholm, SE-106 91, Sweden

*Correspondence to*: Arie Staal (a.staal@uu.nl)

**Abstract.**

Many areas across the globe rely for their precipitation supply on terrestrial precipitation recycling, which is the amount of precipitation that has evaporated from upwind land areas. Global warming and land-use changes may affect the future patterns of terrestrial precipitation recycling, but where and to which extent remains unclear. To study how the global patterns of precipitation recycling may change until the end of the 21st century we present a new forward-tracking version of the three-
dimensional atmospheric moisture tracking model UTrack that is forced by output of the Norwegian Earth System model (NorESM2). We simulate global precipitation recycling in four Shared Socioeconomic Pathways (SSPs), which are internally consistent combinations of climate- and land-use scenarios used in the sixth phase of the Coupled Model Intercomparison Project. The scenarios range from mild to severe: SSP1-2.6, SSP2-4.5, SSP3-7.0, and SSP5-8.5. We compare results for the middle of the century (2050–2059) and end of the century (2090–2099) with a 2015–2024 baseline. We similarly also calculate
basin precipitation recycling for the 26 major river basins of the world. We find that the global terrestrial precipitation recycling ratio decreases with the severity of the SSPs and estimate a decrease in this ratio of 2.1% with every degree of global warming. However, we find differences among regions and river basins in trends in precipitation recycling and whether projected drying or wetting is mainly contributed by land or ocean. Our results give critical insight into the relative contributions of global warming and land use changes on global precipitation changes over the course of this century. In addition, our model paves
the way for more detailed regional studies of future changes in terrestrial moisture recycling.

## 1 Introduction

The global water cycle is a key component of the Earth system that shapes biome distributions, determines vegetation and agricultural productivity, modifies climates, and redistributes energy globally (Gleeson et al., 2020). Water is transported through the atmosphere from the oceans to the continents, where it may precipitate and be used by plants for transpiration or
by humans for agriculture or other purposes. Water that subsequently evaporates or transpires from the land may reprecipitate over land, a phenomenon that is called terrestrial moisture recycling (Van der Ent et al., 2010). Even though eventually all water will return to the oceans either through the atmosphere or as runoff via rivers, recycling over land plays a large role in global precipitation patterns. Roughly half of all current precipitation over Earth's land surface originated as evapotranspiration from land, which amounts to 70% of terrestrial evapotranspiration returning over land (Tuinenburg et al., 2020). Across and
within continents, however, large differences exist in the levels of terrestrial recycling of evapotranspiration and precipitation, depending on, among other factors, the area and size of the continents, the climate, the dominant wind directions, and land cover. For example, the terrestrial precipitation recycling ratio (the fraction of precipitation originating from land) approaches one in eastern Eurasia due to the continental positioning and the westerlies, but it is similarly high in parts of South America and Africa (Van der Ent et al., 2010), which is partially attributed to the moisture recycling capacities of their major tropical
rainforests (Spracklen et al., 2018).

Since pre-industrial times, the global water cycle has undergone considerable changes (Porkka et al., 2024), mainly due to global climate change and land-use changes. These drivers will almost certainly continue to change during the course of this century (IPCC, 2021). Global climate change will cause warming of the atmosphere as well as changes in atmospheric circulations (IPCC, 2021). It is expected that, on average, global warming will decrease terrestrial precipitation recycling ratios (Findell et al., 2019). However, regional differences in changes in land and sea sources of precipitation are likely (Fernández-Alvarez et al., 2023), as well as differences among seasons (Benedict et al., 2019; Fernández-Alvarez et al., 2023). Land-use changes also affect the water cycle, for instance because land-cover type exerts a major influence on the rate and timing of evapotranspiration (Gordon et al., 2005; Sterling et al., 2013). In particular, forests redistribute water more intensively than other natural ecosystems during at least part of the year. Therefore, in recent decades, forest loss in deforestation frontiers like the Amazon has had regionally drying effects (Staal et al., 2020a) whereas increases in leaf area in other parts of the globe have had regionally wetting effects (Cui et al., 2022). Also different human-dominated land-cover types have different effects on the water cycle, for instance in case of rainfed versus irrigated agriculture (Bosmans et al., 2017). However, where, how, and to which extent terrestrial moisture recycling will change in the future remains unclear.

It is generally expected that climate change and land-use changes will develop in tandem over the course of this century. For example, meeting the goals of the Paris Agreement requires both large reductions in carbon emissions and active drawdown of carbon from the atmosphere (Schleussner et al., 2016), which likely involves ecosystem restoration globally (Deng et al., 2023). Such mutually consistent scenarios of climate change and land-use changes for the 21$^{st}$ century are provided by the Shared Socioeconomic Pathways (SSPs) (Riahi et al., 2017). The SSPs provide a framework of five different narratives involving varying degrees of challenges associated with mitigation or adaptation. From each narrative follow different implications for greenhouse gas emissions, energy, and land use. The SSPs serve as the conceptual framework behind the sixth generation of the Coupled Model Intercomparison Project, CMIP6 (Eyring et al., 2016). CMIP6 endorses different specific model intercomparison projects (MIPs) which address different science questions and "grand challenges". ScenarioMIP addresses the long-term (up to 2100) response of the climate system to the SSPs and prioritizes the challenge related to changing water availability. As such, ScenarioMIP provides a suitable conceptual basis for assessing changes in moisture recycling under various futures. The SSPs are combined with a projected radiative forcing level by the year 2100, ranging from 1.9 W/m$^2$ to 8.5 W/m$^2$ (O'Neill et al., 2016). The radiative forcing levels are based on the RCPs (representative concentration pathways) (Van Vuuren et al., 2011). The SSPs and RCPs were combined to create a matrix of possible scenarios. However, not every forcing level coincides with an SSP, so four combinations in this framework serve as "Tier 1 scenarios", which are to be used in all of the models that are a part of ScenarioMIP, ranging from a scenario of sustainable development with 2.6 W/m$^2$ radiative forcing by 2100 to a fossil-fuel-developed global capitalist economy with 8.5 W/m$^2$ radiative forcing: SSP1-2.6, SSP2-4.5, SSP3-7.0, and SSP5-8.5.

The lack of understanding of the future water cycle exists not only because of fundamental uncertainty about the future developments of the global climate and global land-cover distributions, as reflected in the SSPs, but also due to a lack of tools to assess changes in terrestrial moisture recycling in response to these drivers. Terrestrial moisture recycling is often assessed using so-called atmospheric moisture tracking models. These models are used to study moisture recycling on different scales and with different purposes: for example, the upwind water supplies of cities (Keys et al., 2018), the effects of land cover changes on global breadbaskets (Bagley et al., 2012), the role of tropical forests in maintaining their own rainfall levels (Staal et al., 2020b), and to assess global patterns of continental recycling (Van der Ent et al., 2010). Moisture tracking models generally use atmospheric reanalysis data of wind speed and direction, atmospheric moisture content, and evapotranspiration and precipitation to simulate and thereby reconstruct atmospheric moisture flows. This is done either forward in time, tracking

moisture from its evapotranspiration origins to precipitation destinations, or backward in time, tracking moisture from precipitation destinations to evapotranspiration origins. In the models, either the globe is divided into grid cells between which a certain amount of moisture flows at every time step (Eulerian models, e.g. Van der Ent et al., 2014) or individual parcels are tracked through space, in which their coordinates and moisture content are updated at every time step (Lagrangian models). An example of the latter type of model is UTrack (Tuinenburg and Staal, 2020), which is used to track moisture at high spatial and temporal detail through three-dimensional space. In its default form, UTrack uses hourly data for 25 atmospheric layers at 0.25° horizontal resolution from the ERA5 reanalysis dataset (Hersbach et al., 2020). Building upon the methods and principles of the default version of UTrack, we present a new model version that is forced by ScenarioMIP output to study how terrestrial moisture recycling may change across the globe over the course of the 21st century.

## 2 Methods

### 2.1 Lagrangian moisture tracking with UTrack

UTrack is a Lagrangian atmospheric moisture tracking model, which tracks moisture either forward in time (from evaporation to precipitation) or backward in time (from precipitation to evaporation). It tracks the three-dimensional atmospheric trajectories of large numbers of "parcels" of moisture, where the coordinates of each parcel are updated every time step. The number of parcels that is tracked from a certain area and time step depends on the evaporation (in case of forward tracking) or precipitation (in case of backward tracking) from the respective location or area at the respective time step. In the original model version (Tuinenburg and Staal, 2020), the trajectories of the parcels are forced by ERA5 reanalysis data (Hersbach et al., 2020). These forcing data consist of global hourly wind speed and direction for 25 pressure layers at 0.25° horizontal resolution, moisture profiles along the atmospheric column, evaporation, total precipitable water, and precipitation for each grid cell of 0.25°. At every time step, a number of parcels per mm evaporation or precipitation (to be defined by the user) is released at random locations above the starting area. Each parcel is tracked individually based on the wind speeds and directions at the respective moment and three-dimensional location and its coordinates are updated every time step. In addition, to account for small-scale atmospheric dynamics that increase the vertical mixing of moisture but that are poorly captured by coarse atmospheric data, every parcel has a certain probability at every time step of being reassigned a new vertical position. This is done such that a parcel will be repositioned on average once every 24 hours, where the probability of the new position scales with the moisture content along the atmospheric column. Not only the positions, but also the moisture content of the parcels is updated if precipitation (in case of forward tracking, otherwise evaporation) occurs at that time step and location. The amount that rains out from the parcel is equal to the amount of tracked moisture that is still present in the parcel times the ratio of precipitation over the total precipitable water along the atmospheric column. This moisture is then allocated to the grid cell above which the parcel resides. The tracking and updating continues until 99% of the original moisture in the parcel has been allocated or if 30 days have passed since tracking started (whichever comes first). For equations of the moisture tracking model, we refer to Tuinenburg & Staal (2020) and Tuinenburg et al. (2020).

### 2.2 Forcing data

Here, ERA5 data are replaced by output from ScenarioMIP, from which we chose the model that produces the most suitable forcing data for UTrack and our purposes. The variables of CMIP6 models are standardized, but a model can nonetheless produce outcomes for any of over one thousand variables. These variables are stored in the database of the Earth System Grid Federation (ESGF), which we scanned based on our requirements. The required variables were: evspsbl (evaporation including sublimation and transpiration), pr (precipitation), prw (precipitable water), and hus (specific humidity), ua (eastward wind), and va (northward wind) at multiple vertical pressure levels. Furthermore, we desired a temporal resolution not coarser than a day and a high spatial resolution. The only model that met these requirements for all Tier 1 scenarios in ScenarioMIP up until

2100 is the medium-resolution Norwegian Earth System Model version 2, or NorESM2-MM (Seland et al., 2020). The output

of the model, hereafter Nor-ESM2, has a temporal resolution of one day and a spatial (zonal × meridional) resolution of 1.25° × 0.9375°. The wind speeds are calculated for eight pressure levels: 1000 hPa, 850 hPa, 700 hPa, 500 hPa, 250 hPa, 100 hPa, 50 hPa, and 10 hPa. Horizontal fluxes are purely based on these pressure levels, whereas vertical fluxes additionally include the above-mentioned probabilistic parcel repositioning every 24 hours (Tuinenburg and Staal, 2020). Sensitivity tests done by Tuinenburg & Staal (2020) indicate that degrading the vertical moisture profile (here from 25 to eight) can affect moisture

transport distances in the order of hundreds of km, but the number of pressure levels used here is still relatively large. NorESM2 outputs for SSP1-2.6, SSP2-4.5, SSP3-7.0, and SSP5-8.5 are available for the period 2015–2100 (Seland et al., 2020).

NorESM2 is based on the Community Earth System Model (CESM2.1) structure (Danabasoglu et al., 2020), but with modified components (Seland et al., 2020). The land and vegetation component is based on the Community Land Model version 5

(CLM5; Lawrence et al., 2019). The model includes terrestrial ecosystem interactions that drive weather and climate, as well as the land interface to critical climate, social, and ecosystem interactions that influence global environmental changes. The CLM5 land components forced in NorESM2 have an improved land unit weighing system that allows for mechanistic treatment of key processes (soil and plant hydrology, snow density, river modeling, carbon and nitrogen cycling and coupling, and crop modeling) as well as comprehensive representation of land and land cover changes. With 64 crop functional types (CFTs) and

15 natural plant functional types (PFTs), the model can represent up to 78 plant functional type distributions represented over a transient of 1850 to 2100 under various climate scenarios (Lawrence et al., 2019).

Historical runs of NorESM2 perform well in reproducing observed levels of global warming and oceanic circulation patterns (Seland et al., 2020). The model also performs relatively well in reproducing multi-annual climatic variability such as the El

Niño Southern Oscillation including El Niño teleconnections (Seland et al., 2020). On reproducing historical observations of the hydrological cycle, the model outperforms other CMIP6 models (Abdelmoaty et al., 2021; Du et al., 2022). Importantly for our purposes, precipitation estimates resemble the observational data across latitudes (Abdelmoaty et al., 2021). In the simulations for the future, it has a relatively low climate sensitivity compared to other CMIP6 models (Seland et al., 2020). However, the model ranks high among the CMIP6 model cohort in terms of simulating future global land precipitation and its

interannual variability (Du et al., 2022). In the CLM5 module, reliably simulating leaf stomatal conductance is key for quantifying effects of environmental perturbations through the land-surface energy, water, and $CO_2$ fluxes, on which it performs highly in predicting observations (Franks et al., 2018; Lawrence et al., 2019).

### 2.3 Simulation settings

We ran the model in time steps of four hours. Although this is coarser than the time step of published UTrack runs using ERA5 data, which is either 0.1 hours (Tuinenburg et al., 2020) or 0.25 hours (Staal et al., 2023), it is six times as fine as the temporal resolution of the NorESM2 forcing data and in this regard comparable to the ERA5-based model versions. We used these forcing data directly without interpolation. Individual moisture parcels may cross multiple grid cells during one time step if the time step is too large. This may cause errors in the parcel trajectories, which is solved by taking a sufficiently small time

step, even if the data themselves are not interpolated. Because NorESM2 produces globally covered precipitation data, but evaporation data only for land areas, only forward tracking from land areas was possible: in order to allocate evaporation from a source area to precipitation, global coverage of precipitation is required; vice versa, in order to allocate precipitation at a sink area to evaporation, global coverage of evaporation is needed. We performed forward tracking from all global land cells. Here, for each mm of globally averaged evaporation during each four-hour time step, we released 1000 moisture parcels. Because

evaporation is not equally distributed across the globe, we assigned a random initial position to each parcel for which the

probability was weighted by the evaporation distribution during the respective time step. We thus ran the model for SSP1-2.6, SSP2-4.5, SSP3-7.0, and SSP5-8.5 and stored the global output for each month. Depending on the scenario and year, we tracked around 320,000 moisture parcels per simulation year and scenario.

In addition to global recycling ratios, we were interested in basin precipitation recycling ratios for individual river basins located across the globe. Analogous to global terrestrial precipitation recycling, we calculated basin precipitation recycling ratios as the percentage of precipitation that originated as evaporation from the same basin. Because the global runs involved simultaneous parcel tracking from everywhere across the globe, which do not allow for calculating basin recycling, we required separate runs for this. Therefore, we performed forward-tracking runs for the 26 major river basins of the world using shapefiles

from the Global Runoff Data Centre (GRDC, 2020). We performed these runs again for all SSPs. Instead of 1000 parcels for every mm of evaporation globally, we released 100 parcels for every mm of evaporation from each basin. Note that this implies a larger amount of parcels per volume than in the global runs due to the considerably smaller source area of the basins.

### 2.4 Analysis

We take the first ten years (2015–2024) from the SSP2-4.5 scenario as a baseline to compare global precipitation recycling (ratios) under future scenarios with, because this SSP represents the middle-of-the-road trajectory that the world is currently on (Fricko et al., 2017; Lee et al., 2023). We calculate the global terrestrial precipitation recycling ratio as the percentage of precipitation on land that evaporated from land. Similarly, we calculate basin precipitation recycling ratios as the percentage of precipitation within a basin that evaporated within the basin. We focus on comparisons between the baseline period and the

middle of the century (2050–2059, figures in the supplement) as well as the end of the century (2090–2099, figures in the main text). These comparisons happen on a per grid-cell basis for each SSP. We check for statistical significance of change using a t-test comparing the annual recycling values in the baseline period with those of the 2050s or 2090s. We also report global evaporation recycling, the percentage of terrestrial evaporation that precipitates over land. Calculating evaporation recycling is possible given that the source area of the tracking equals the target area (the global land area). We calculate global average

changes in precipitation and evaporation recycling ratios for each degree of warming. For this we determined the global near-surface temperature rise for global land in NorESM2 , between the baseline and 2090–2099 in the SSP5-8.5 scenario, which was 4.7 °C.

To better understand whether trends in precipitation are caused by changes in moisture contributions from the ocean or from

the land, we divide the grid cells with significant changes in precipitation (recycling) into four categories: "wetting, land-dominated" if a significant increase in precipitation coincides with a significant ($\alpha = 0.05$) increase in terrestrial precipitation recycling ratio; "wetting, ocean-dominated" if a significant increase in precipitation coincides with a significant decrease in terrestrial precipitation recycling ratio; "drying, ocean-dominated" if a significant decrease in precipitation coincides with a significant decrease in terrestrial precipitation recycling ratio; and "drying, land-dominated" if a significant decrease in

precipitation coincides with a significant increase in terrestrial precipitation recycling ratio. Furthermore, we report the changes in forest cover and cropland cover, including global and river-basin averages (again between the baseline, 2050s, and 2090s for each SSP), and of evaporation and precipitation.

### 2.5 Model evaluation

To evaluate our model results against the literature, we compared precipitation recycling ratios from NorESM2 with UTrack simulations based on ERA5 for 2008–2017 (Tuinenburg and Staal, 2020). The global patterns are qualitatively similar, but the

NorESM2-based estimates in high-latitude boreal zones are relatively low compared to those based on ERA5 (Fig. A5). We used the ERA5-based basin recycling ratios for the 26 major river basins as reported by Tuinenburg et al. (2020) for quantitative comparisons. We performed regressions between these basin recycling ratios and those from our baseline period. The reason we evaluated based on basin recycling rather than grid-cell-by-cell is the different spatial resolution between model versions and expected noise at relatively small spatial scales. The estimates based on NorESM2 are on average 9.4 percentage point lower than those from Tuinenburg et al. (2020). The absolute differences are on average 9.8 percentage point (Fig. A6). The fact that the average absolute differences are very similar to the average differences shows that the bias is systematic; only for two out of 26 river basins, the NorESM2-based estimates are slightly larger. Because we are primarily interested in relative changes in recycling (and the differences among SSPs therein), we believe a systematic bias like this is acceptable for our purposes.

We also evaluated the choice of a ten-year time slice for our analysis. For this, we plotted the moving averages ± one standard deviation of the global terrestrial precipitation recycling ratios based on a ten-year time slice and a 30-year time slice, for each of the SSPs. We found that these ratios and their standard deviations largely overlap, indicating that ten-year time slices tend to be sufficient to capture most of the interannual variability in global precipitation recycling. Furthermore, especially in the severe scenarios SSP3-7.0 and SSP5-8.5 the trend in recycling ratio exceeds its variability, implying that long-term climate variability in NorESM2 does not affect our main outcomes (Fig. A7).

For NorESM2, we have one daily value for the wind field, an instantaneous value at 00Z. This may be a biased value compared to a higher temporal resolution of the daily cycle in wind speed. Wind speeds may be systematically different during different times of the day, which may lead to this bias. Moreover, because 00Z is at different solar (local) times around the globe, these biases may be spatially differing. Therefore, we estimated the bias in wind speed based on the ERA5 atmospheric reanalysis at different times of the day for the period 2010–2023. We retrieved the monthly mean reanalysis by time of day for the variables U and V between 1000 and 500 hPa. We calculated a quasi mean absolute wind speed between 1000-500 hPa based on these monthly values. Note that this is a 'quasi' wind speed, as we use the monthly mean U and V values, which is not the same as the monthly mean absolute wind speed. For both U and V, the hourly values have positive and negative values within a month, which will cancel out and thus not contribute to the absolute wind speed. In Fig. A8 we represent the absolute and relative difference of the 00Z quasi wind speed with the daily mean quasi wind speed. Typically, the absolute wind speed at 00Z deviates less than 0.2 m/s from the daily mean, although there are some regions with deviations up to 1 m/s. In relative terms, this deviation is typically within 5% of the wind speed, but with 10% deviation in some areas. A positive deviation will probably mean that the moisture recycling is underestimated, while a negative deviation will mean that the moisture recycling is overestimated. It is hard to translate these wind deviations to quantitative values of moisture recycling deviations, but given the low relative wind deviations, we expect the moisture recycling uncertainty due to this effect to be relatively small.

## 3 Results

### 3.1 Global terrestrial precipitation recycling changes

Averaged across the globe, both terrestrial precipitation and terrestrial evaporation increase in all scenarios by the middle of the century (2050–2059) and the end of the century (2090–2099), although not significantly in SSP2-4.5 for the middle of the century (Table 1). The largest increase in global precipitation occurs in SSP5-8.5 for the end of the century, from 604 mm year$^{-1}$ in the baseline scenario (SSP2-4.5 for 2015–2024) to 647 mm year$^{-1}$, amounting to a 7% increase globally. This projection is typical for IPCC models, among which the average projected global precipitation increase by the end of the

century is 6.6% (ranging between 3.3–11%) (IPCC, 2021). The largest increase in global evaporation occurs in SSP1-2.6 for the end of the century, from 315 mm year$^{-1}$ in the baseline scenario to 328 mm year$^{-1}$, amounting to a 4% increase globally (Table 1). Forest cover globally is projected to increase in only SSP1-2.6, to 27%, from 25% in the baseline scenario. This increase is reached already by the 2050s, after which no change is projected until the end of the century (Tables A2, A3; Fig. A5). Global cropland cover increases in all scenarios from 11% towards the end of the century, peaking in SSP3-7.0 at 14% and in SSP1-2.6 only after a small decrease by mid-century (Tables A2, A3; Fig. A6).

In the 2015–2024 (SSP2-4.5) baseline period, the global terrestrial precipitation recycling ratio is 34.0% (± 0.37% annual standard deviation) (Fig. 1, Table 1). This ratio does not change significantly for 2090–2099 in SSP1-2.6 (34.3% ± 0.58%). It does decrease significantly (p < 0.01) to 33.4% (± 0.39%) in SSP2-4.5; to 32.5% (± 0.42%) in SSP3-7.0 (p << 0.01); and to 31.7% (± 3.8%) in SSP5-8.5 (p ≈ 0) (Fig. 2; Table 1). The decline in global precipitation recycling ratio between the baseline period and SSP5-8.5 for the end of the century is 6.8%. Given a global temperature rise of 4.7 °C in NorESM2 in this scenario, globally averaged precipitation recycling ratio is thus projected to decrease by 1.5% with each degree of warming.

In the baseline period, the global evaporation recycling ratio is 65.2% (± 0.65%). Also this ratio does not change significantly in SSP1-2.6 (65.4% ± 0.54%), but does decrease significantly to 64.3% (± 0.93%) for 2090–2099 in SSP2-4.5 (p = 0.02); to 63.9% (± 0.82%) in SSP3-7.0 (p = 0.01); and to 62.9% (± 0.63%) in SSP5-8.5 (p << 0.01) (Table 1). The decline in global evaporation recycling ratio between the baseline period and SSP5-8.5 for the end of the century is 3.5%. Given the global temperature rise of 4.7 °C, globally averaged evaporation recycling ratio is thus projected to decrease by 0.8% with each degree of warming.

With a more severe SSP, the proportion of global land that experiences a significant change in precipitation by the 2090s increases, from 8.7% of global land cells in SSP1-2.6 to 41.5% in SSP5-8.5. Drying is mostly concentrated in the Amazon and eastern Europe; wetting occurs mostly in the high northern latitudes and in eastern Asia. Whether this change in precipitation is drying or wetting, we find a larger proportion of land grid cells in which terrestrial precipitation recycling ratio decreases as the SSP becomes more severe (Table A4). In SSP5-8.5, 41.5% of all land grid cells show a significant change in precipitation between the baseline period and the end of the century, of which 19.0% are projected to become drier and 81.0% to become wetter (Fig. A1). In 75.5% of the land grid cells that are projected to become drier (representing 6.0% of all land grid cells), this drying is dominated by a decrease in the precipitation that originates from land. In the remaining 24.5% of drying land grid cells (1.9% of all land grid cells), the drying is dominated by a decrease in the precipitation from ocean (see Fig. 3, including non-significant changes). In 32.9% of the land grid cells that are projected to become wetter (11.0% of all land grid cells), this wetting is dominated by an increase in the precipitation originating from land. In the remaining 67.1% of wetting land grid cells (23.0% of all land grid cells), the wetting is dominated by an increase in the precipitation from ocean (Fig. 4, Table A4).

We can look at the robustness across scenarios of the projections of terrestrial precipitation recycling change (Fig. 5). In 20.2% of global land grid cells excluding Antarctica, terrestrial precipitation recycling ratio decreases in all four scenarios (for absolute recycling, in mm year$^{-1}$, this is 12.7% of land grid cells). In 18.7% of global land grid cells, terrestrial precipitation recycling ratio decreases in three, but increases in one scenario (for absolute recycling 11.5%). In 14.2% of land grid cells, terrestrial precipitation recycling is projected to decrease in two and increase in two scenarios (for absolute recycling (11.3%). In 11.8%, an increase is projected in three and a decrease in the remaining scenario (for absolute recycling 13.5%). Finally, in 12.1% of global land grid cells, terrestrial precipitation recycling is projected to increase in all scenarios (for absolute recycling 28.1%) (Fig. 5).

We observe considerable seasonality in the future terrestrial precipitation recycling change in SSP2-4.5 (Fig. 6). This seasonality coincides with shifts in the belt of low pressure near the equator where the trade winds of the Northern Hemisphere and Southern Hemisphere converge, called the Intertropical Convergence Zone (ITCZ). The positioning of the ITCZ makes the seasonality of moisture flow very variable based on the timing of the wet season: north of the equator mostly between June and August, and south of the equator mostly between December and February. We see a large-scale and strong reduction in recycling ratio during June–August south of the equator, in both South America and Africa (Fig. 6). In the temperate and boreal Northern Hemisphere, changes in terrestrial recycling ratio are most pronounced between March and August, the growing season.

In some grid cells, estimated precipitation recycling exceeds precipitation itself due to model artifacts. In the baseline scenario, this occurs in 1.3% of global land grid cells. These areas are depicted as having a precipitation recycling ratio of 100% in Fig. 1a and are mainly located in the Himalaya and the Andes mountains. Sometimes, too many forward-tracked moisture parcels end up in a grid cell relative to the precipitation in that grid cell in that month. This can be due to the stochastic nature of the model, the fact that parcels can be tracked across two months with a different water balance. The area where recycling exceeds actual precipitation remains stable across scenarios: for 2090–2099, in SSP1-2.6, this occurs in 1.3% of global land grid cells; in SSP2-4.5, in 1.2% of global land grid cells; in SSP3-7.0, in 1.1% of global land grid cells; and in SSP5-8.5, in 1.0% of global land grid cells.

### 3.2 Precipitation recycling changes for the major river basins

As the scenario becomes more severe, a larger proportion of the 26 major river basins of the world is projected to undergo significant changes in basin precipitation recycling ratio and terrestrial precipitation recycling ratio. In SSP1-2.6, by the end of the century, two basins have a statistically significant change in basin recycling and also two in terrestrial recycling at the $\alpha = 0.05$ level. In SSP2-4.5, five basins have a significant change in basin recycling and four in terrestrial recycling ratio. In SSP3-7.0, this increases to seventeen basins with a change in basin recycling and seven with a change in terrestrial recycling. In SSP5-8.5, we find the largest number of significant changes, with nineteen changes in basin recycling and ten in terrestrial recycling (Table 2).

The great majority of significant changes in either basin or terrestrial precipitation recycling ratio is a decrease (Table 2). The only scenarios in which an increase (with at least a 1% increase in rounded values) occurs is in SSP1-2.6 and SSP2-4.5. The two basins that showed statistically significant changes in basin recycling ratio by the end of the century in SSP1-2.6 both showed increases from the baseline. These basins are the Amur basin (from 19% to 20%), within which an increase in forest cover (from 43% to 47%) and a decrease in crop cover (from 10% to 9%) are projected, and the Ob basin (from 11% to 13%), within which an increase in forest cover (from 33% to 35%) but no change in crop cover (15%) is projected (Table A3). Also both changes in terrestrial recycling are an increase, again for the Ob basin (from 45% to 48%) as well as for the Yenisey basin (from 57% to 60%). The only increases in recycling in SSP2-4.5 are in terrestrial precipitation recycling ratio for the Indus basin (from 73% to 90%) and the Kolyma basin (from 33% to 36%) (Table 2).

Both in an absolute and relative sense, the largest projected decrease in basin precipitation recycling ratio among all scenarios is for the Orange basin in SSP5-8.5, from 16% to 12%, amounting to a 24% decrease. The largest projected absolute decrease in terrestrial recycling ratio among all scenarios is for the Nelson basin in SSP5-8.5, from 50% to 43%. The largest projected relative terrestrial recycling ratio decrease is for the Mississippi basin in SSP5-8.5, from 42% to 36%, amounting to a 15% decrease (Table 2).

Basin precipitation recycling ratios by the middle of the century (2050–2059) tend to be larger than those by end of the century, but not always, especially in the milder scenarios (cf. Tables 2, A1). In SSP1-2.6, there are seven basins that have an increase in basin recycling ratio between the middle and end of the century, which are the Chad, Euphrates-Tigris, Mackenzie, Mississippi, Nile, Ob, and Yukon basins. In SSP2-4.5, there are five with an increase between the middle and end of the century, which are the Huang He (Yellow River), Mackenzie, Murray, Nelson, and Orange basins. In SSP3-7.0 and SSP5-8.5, there are no basins with an increase (of at least one percentage point) in basin recycling ratio during the second half of the century. In SSP2-4.5, decreasing basin recycling after the 2050s is projected for the Amazon, Amur, Danube, Euphrates-Tigris, and Indus basins. In SSP3-7.0 and SSP5-8.5, the majority of the 26 major river basins have a decrease in basin recycling ratio after the 2050s: fifteen in SSP3-7.0 and nineteen in SSP5-8.5 (Tables 2, A1).

Generally, climate and land use change simultaneously in individual basins, which tend to show some change in land cover (forest cover and cropland cover) among scenarios and between the middle and the end of the century (Tables A2, A3). Sometimes, however, both forest cover and cropland cover remain equal, while basin recycling ratios do change. This gives an indication of the effect of only climate change alone on moisture recycling. For instance, in the Amazon, in both the baseline scenario and in SSP5-8.5 (both 2050–2059 and 2090–2099), forest cover is 82% and cropland cover is 3% (Table A3) whereas the basin recycling ratio decreases significantly from 27% to 25% to 24% (Tables 2, A1). This can be explained by increased residence times of moisture in a warmer atmosphere, increasing the typical distance that moisture travels before precipitating. The other basins with equal forest cover and cropland cover in both the baseline scenario and SSP5-8.5 (2090–2099) are the Lena basin (54% forest and 0% cropland), Mackenzie basin (46% forest and 2% cropland), and Yukon basin (36% forest and 2% cropland) (Table A3). In all cases, the basin recycling ratio decreases significantly, respectively from 16% to 13%, from 16% to 13%, and from 8% to 6% (Table 2). The Huang He (Yellow River) basin has 7% forest cover and 15% cropland cover in both the baseline scenario and in SSP2-4.5 (2090–2099) (Table A3), whereas the basin recycling ratio decreases significantly from 19% to 18% (Table 2).

In contrast to differences in recycling ratios without differences in land cover, end-of-century recycling ratios are sometimes the same between SSPs with different land cover distributions, which provides useful information about the isolated effect of land-cover change. For instance, in the Amazon, under SSP3-7.0 and SSP5-8.5 the estimated basin recycling ratio is 24% (Table 2), despite a forest cover of 75% in SSP3-7.0 and 82% in SSP5-8.5 (Table A3). Thus, the decrease in recycling ratio due to a warmer from SSP3-7.0 to SSP5-8.5 equals the increase in recycling ratio due to seven percentage point larger forest cover. Similarly in the Congo, under SSP3-7.0 and SSP5-8.5 the basin recycling ratio is 30% and the terrestrial recycling ratio is 52% (Table 1), with a large difference in forest cover: 27% in SSP3-7.0 and 50% in SSP5-8.5 (Table A3). Here, the decrease in recycling ratio due to a warmer climate from SSP3-7.0 to SSP5-8.5 equals the increase in recycling ratio due to 23 percentage point larger forest cover.

**Table 1.** Global terrestrial precipitation (mm year$^{-1}$), terrestrial precipitation recycling ratio (%), standard deviation of terrestrial precipitation recycling ratio (%), minimum and maximum terrestrial precipitation recycling ratio (%), terrestrial evaporation (mm year$^{-1}$), terrestrial evaporation recycling ratio (%), and standard deviation of terrestrial evaporation recycling ratio (%), and minimum and maximum terrestrial evaporation recycling ratio (%),  in the baseline scenario (SSP2-4.5 during 2015–2024) and the end of the century (2090–2099) in SSP1-2.6, SSP2-4.5. SSP3-7.0, and SSP5-8.5. Significant differences with the baseline are indicated by one (for p < 0.05) or two (for p < 0.01) asterisks.

| | Baseline | SSP1-2.6 2090s | SSP2-4.5 2090s | SSP3-7.0 2090s | SSP5-8.5 2090s |
|---|---|---|---|---|---|
| Precipitation (mm year$^{-1}$) | 604 | 626** | 623* | 636** | 647** |
| Precipitation recycling (%) | 34.0 | 34.3 | 33.4** | 32.5** | 31.7** |
| St. dev. precipitation recycling (%) | 0.37 | 0.58 | 0.39 | 0.42 | 0.38 |
| Min. precipitation recycling (%) | 33.6 | 33.7 | 32.7 | 31.8 | 31.2 |
| Max. precipitation recycling (%) | 34.7 | 35.8 | 34.0 | 33.3 | 32.6 |
| Evaporation (mm year$^{-1}$) | 315 | 328** | 323* | 324** | 326** |
| Evaporation recycling (%) | 65.2 | 65.4 | 64.3* | 63.9* | 62.9** |
| St. dev. evaporation recycling (%) | 0.65 | 0.54 | 0.93 | 0.82 | 0.63 |
| Min. evaporation recycling (%) | 64.8 | 64.5 | 62.9 | 62.3 | 61.8 |
| Max. evaporation recycling (%) | 66.8 | 66.3 | 65.6 | 64.9 | 63.9 |

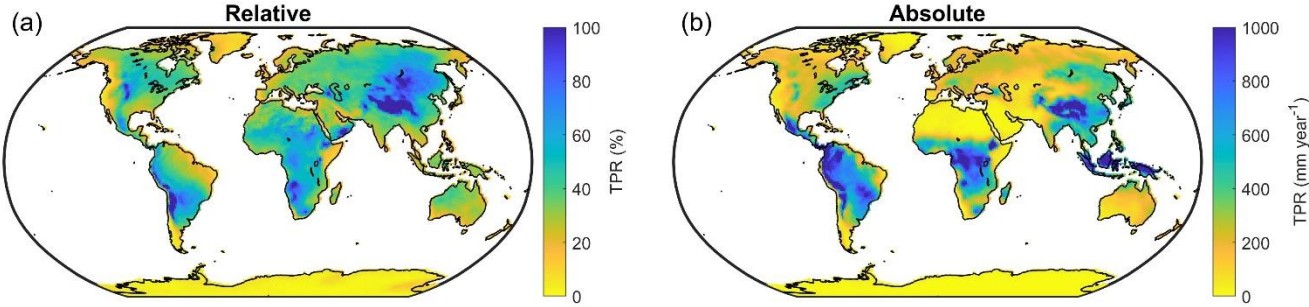

**Figure 1.** Terrestrial precipitation recycling across the globe in the baseline scenario (SSP2-4.5 for 2015–2024); (a) Relative terrestrial moisture recycling ratio in percent; (b) Absolute terrestrial moisture recycling in mm/year. Note that the color scale in (b) is truncated at 1000 mm/year.

380

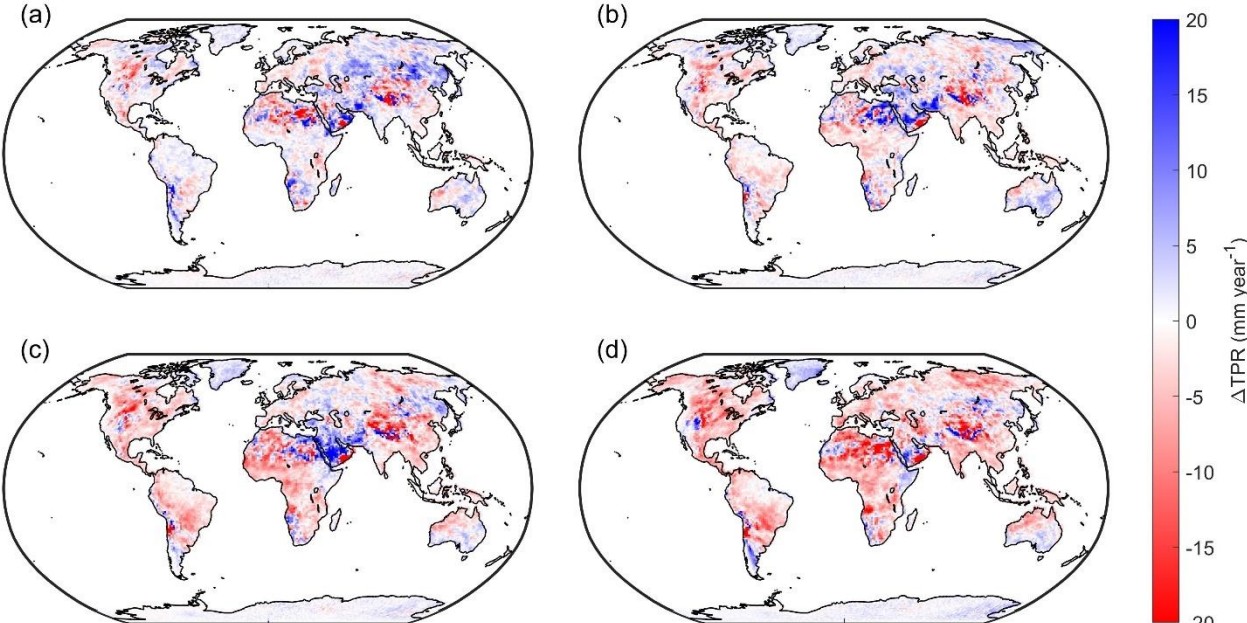

**Figure 2.** Differences in relative annual terrestrial precipitation recycling ratio (ΔTPR) across the globe between the baseline period (2015–2024) and the end of the century (2090–2099) in percentage points, for (a) SSP1-2.6, (b) SSP2-4.5, (c) SSP3-7.0, and (d) SSP5-8.5. Positive values indicate an increase in precipitation recycling ratio and negative values a decrease. Both significant and non-significant differences are shown. Color scales are truncated at -20 percentage points and 20 percentage points.

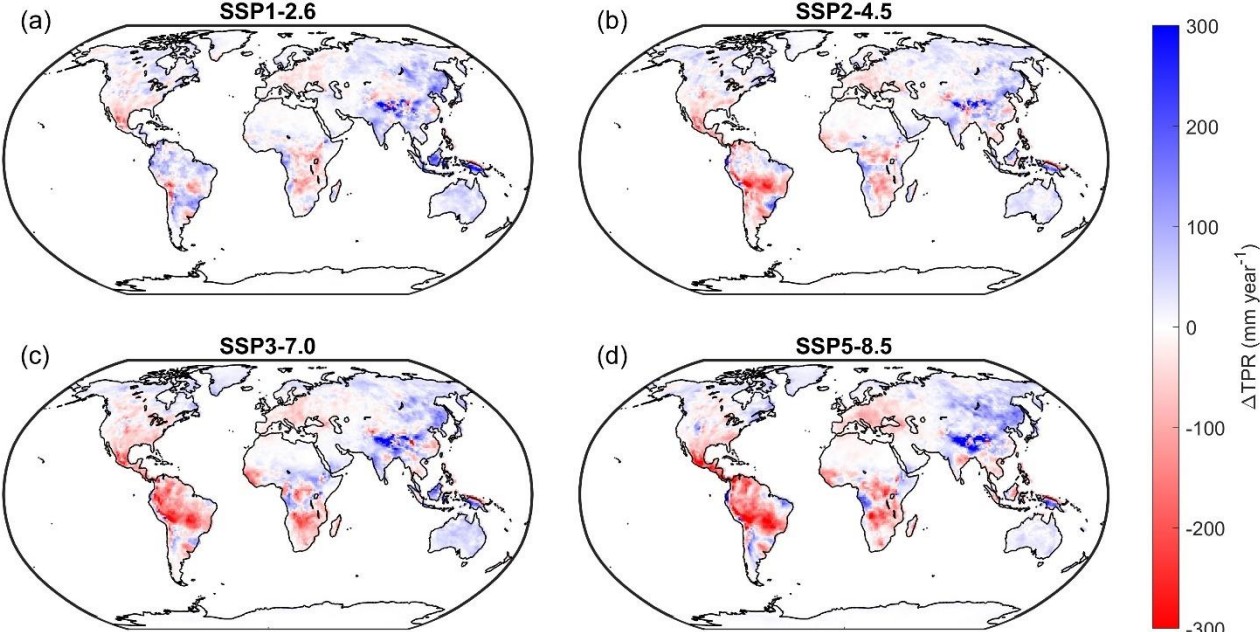

**Figure 3.** Differences in absolute annual terrestrial precipitation recycling (ΔTPR) across the globe between the baseline period (2015–2024) and the end of the century (2090–2099) in mm year$^{-1}$, for (a) SSP1-2.6, (b) SSP2-4.5, (c) SSP3-7.0, and (d) SSP5-8.5. Positive values indicate an increase in precipitation recycling and negative values a decrease. Both significant and non-significant differences are shown (but also see Fig. 4). Color scales are truncated at -300 mm year$^{-1}$ and 300 mm year$^{-1}$.

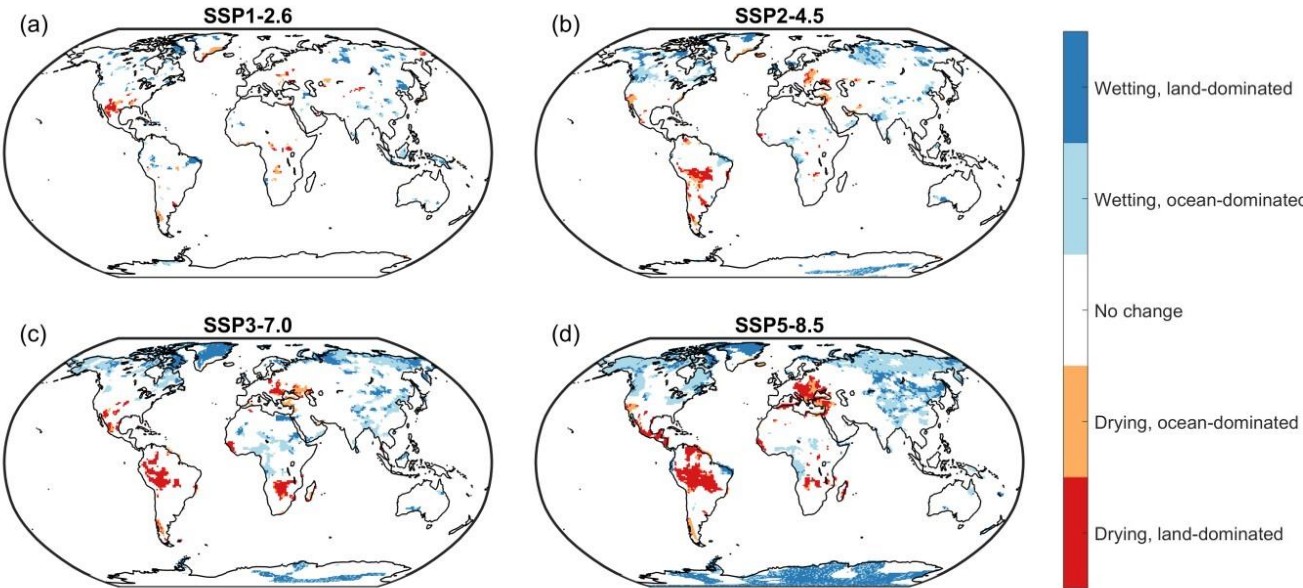

**Figure 4.** Wetting (in blue) and drying (in red) areas across the globe between the baseline period (2015–2024) and the end of the century (2090–2099) dominated either by changes in precipitation originating from land (darker colors) or from ocean (lighter colors), for (a) SSP1-2.6, (b) SSP2-4.5, (c) SSP3-7.0, and (d) SSP5-8.5. Non-significant changes in annual precipitation are shown in white.

400

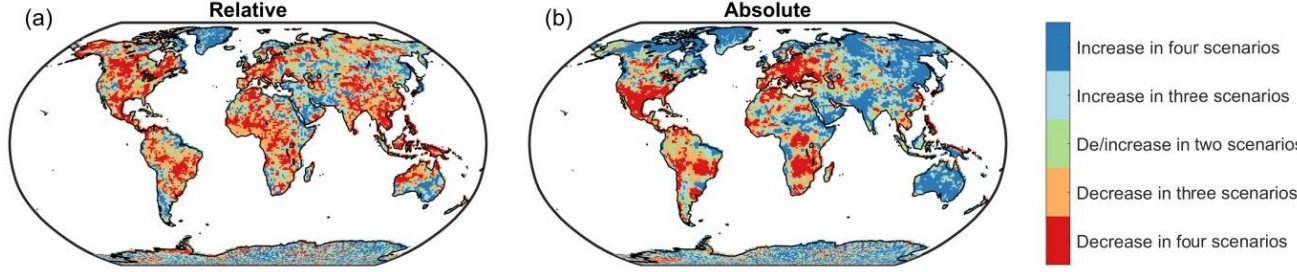

**Figure 5.** Robustness of projections across scenarios of terrestrial precipitation recycling by the end of the century (2090–2099). (a) Relative precipitation recycling (%); (b) Absolute precipitation recycling (mm year$^{-1}$). Both significant and non-significant changes in recycling are included.

405

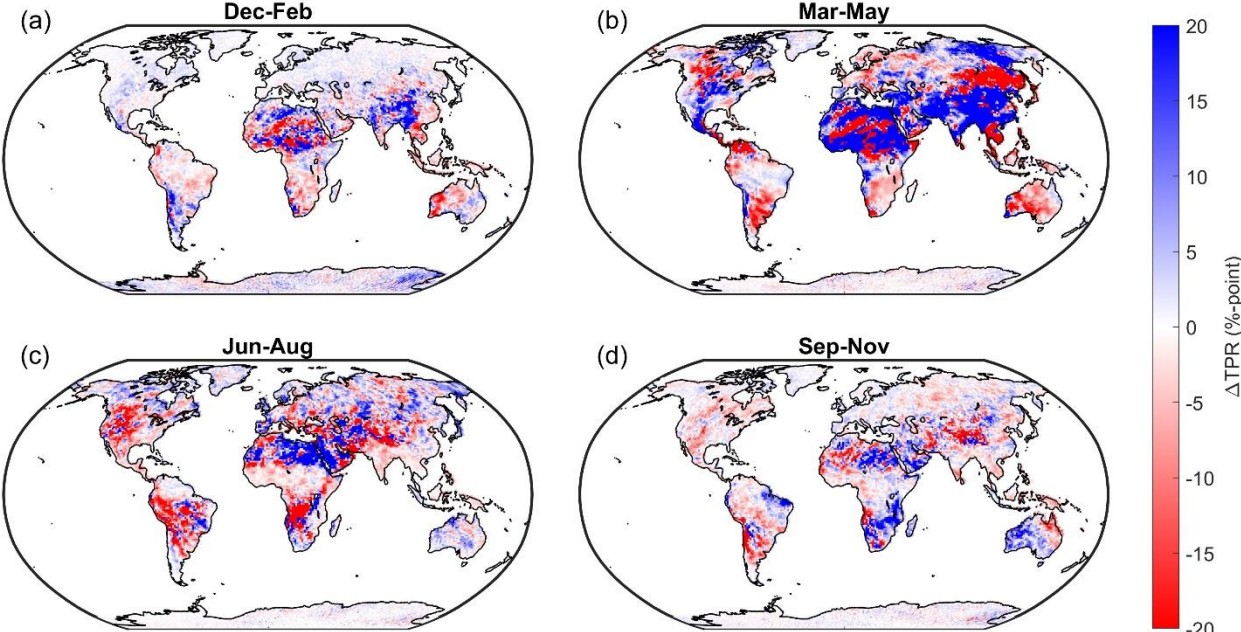

**Figure 6.** Differences in monthly terrestrial precipitation recycling ratio (ΔTPR) across the globe between the baseline period (2015–2024) and the end of the century (2090–2099) for SSP2-4.5 in percentage point, for (a) December–February, (b) March–May, (c) June–August, and (d) September–November. Positive values indicate an increase in precipitation recycling and negative values a decrease. Both significant and non-significant differences are shown. Color scales are truncated at -20 percentage points and 20 percentage points.

410

415 **Table 2.** Basin and terrestrial precipitation recycling ratios (%) for the 26 major river basins of the world in the baseline scenario (SSP2-4.5 during 2015–2024) and by the end of the century (2090–2099) in SSP1-2.6, SSP2-4.5. SSP3-7.0, and SSP5-8.5. Basin recycling ratio is the percentage of precipitation in a basin that has evaporated from the same basin; terrestrial precipitation recycling ratio is the percentage of precipitation on land that has evaporated from land. For results for 2050–2059, see Supplementary Table A1. Significant differences with the baseline are indicated by one (for $p < 0.05$) or two (for $p < 0.01$) asterisks.

| | Baseline | | SSP1-2.6 | | SSP2-4.5 | | SSP3-7.0 | | SSP5-8.5 | |
|---|---|---|---|---|---|---|---|---|---|---|
| | Basin | Terr. | Basin | Terr. | Basin | Terr. | Basin | Terr. | Basin | Terr. |
| Amazon | 27 | 43 | 26 | 43 | 25** | 41** | 24** | 40 | 24** | 40* |
| Amur | 19 | 65 | 20* | 71 | 18 | 65 | 18 | 68 | 17 | 66 |
| Chad | 16 | 59 | 15 | 58 | 14 | 56 | 14* | 54 | 13** | 52 |
| Congo | 33 | 56 | 33 | 56 | 32* | 55* | 30** | 52* | 30** | 52** |
| Danube | 9 | 38 | 9 | 37 | 9 | 36 | 8 | 35 | 8** | 33** |
| Euphrates-Tigris | 8 | 35 | 9 | 37 | 8 | 41 | 8 | 38 | 7 | 33 |
| Ganges | 9 | 46 | 9 | 45 | 9 | 43 | 8** | 44 | 7** | 42 |
| Huang He | 19 | 82 | 18 | 78 | 18* | 79 | 16** | 77 | 15** | 79 |
| Indus | 16 | 73 | 17 | 87 | 17 | 90* | 16 | 84 | 16 | 86 |
| Kolyma | 5 | 33 | 5 | 36 | 5 | 37* | 5 | 33 | 4 | 31 |
| Lena | 16 | 59 | 16 | 60 | 15 | 57 | 15 | 57 | 13** | 54 |
| Mackenzie | 16 | 39 | 16 | 39 | 16 | 38 | 13* | 36 | 13** | 34* |
| Mississippi | 15 | 42 | 16 | 42 | 15 | 40 | 14* | 37* | 13** | 36** |
| Murray | 9 | 28 | 9 | 28 | 10 | 29 | 9 | 29 | 9 | 29 |
| Nelson | 10 | 50 | 10 | 47 | 10 | 46 | 9** | 44** | 8** | 43** |
| Niger | 11 | 45 | 11 | 45 | 10* | 43 | 10** | 40* | 9** | 40 |
| Nile | 24 | 57 | 24 | 58 | 23 | 57 | 22* | 56 | 22* | 54 |
| Ob | 11 | 45 | 13* | 48* | 11 | 44 | 10 | 44 | 11 | 44 |
| Orange | 16 | 62 | 14 | 59 | 15 | 60 | 14* | 60 | 12** | 55 |
| Paraná | 24 | 66 | 24 | 66 | 23* | 65 | 22** | 62* | 21** | 60* |
| Saint Lawrence | 8 | 43 | 9 | 42 | 8 | 40 | 8** | 38* | 8* | 38** |
| Volga | 8 | 39 | 8 | 39 | 8 | 39 | 8 | 41 | 8 | 37 |
| Yangtze | 19 | 75 | 19 | 73 | 19 | 76 | 17* | 70 | 17** | 71 |
| Yenisey | 14 | 57 | 14 | 60* | 13 | 56 | 13* | 56 | 13* | 56 |
| Yukon | 8 | 27 | 8 | 27 | 7 | 27 | 6* | 24* | 6* | 24* |
| Zambezi | 15 | 55 | 16 | 54 | 15 | 53 | 14** | 50 | 14** | 50** |

420

## 4 Discussion

### 4.1 Trends in global moisture recycling

We find that across the 21st century, global terrestrial moisture recycling decreases with the severity of the Shared Socioeconomic Pathways (SSPs). Because these SSPs represent internally consistent scenarios of both global warming and global land cover changes, it is hard to distinguish the relative contributions of these two. However, the evidence points at a dominance of global warming on this result. Whereas global warming increases monotonically with the severity of the scenarios (from 2.6 to 4.5, 7.0, and 8.5 W m$^{-2}$ radiative forcing), land cover change does not. The highest levels of deforestation and cropland conversion occur in SSP3, which is a scenario in which globalisation is reversed and global inequality is large. This leads to significant increases in forest loss, especially in the tropics. In the more globalised world of SSP5, deforestation occurs considerably less (Riahi et al., 2017).

A warmer atmosphere can hold more moisture, and this increase in atmospheric moisture rises faster than associated increases in precipitation itself (Trenberth et al., 2003). Therefore, as the atmosphere warms, more evaporation from the oceans results in higher atmospheric moisture content (Held and Soden, 2006; O'Gorman and Muller, 2010), which will subsequently be carried by the winds towards the continents, a trend which has indeed been reported for the recent decades (Wang et al., 2023). As a consequence, the global terrestrial precipitation recycling ratio has decreased (Gimeno et al., 2020). Related, if a warmer atmosphere holds more moisture, the same amount of atmospheric moisture resulting from (local) evapotranspiration will represent a smaller proportion of total (local) atmospheric moisture. Therefore, residence times of moisture in the atmosphere will become larger, as well as the typical distance that this moisture will travel before raining down (Gimeno et al., 2021). Our projection of decreasing global terrestrial precipitation recycling by 1.5% with every degree of global warming is slightly lower than estimates from the literature. A 2–3% decrease in global precipitation recycling was found with every degree of warming in an Earth System Model (2.0% °C$^{-1}$ for the 21st century) (Findell et al., 2019) and recent global declines in terrestrial recycling are estimated to have been 1.6% with every degree (Gimeno et al., 2020).

Future work should use different methods to assess the relative contributions of climate change and land cover changes to global moisture recycling. In theory, evaporation over ocean should increase more rapidly than terrestrial evaporation due to their terrestrial soil moisture limitations (Findell et al., 2019). The increase in moisture input from the ocean will follow the Clausius-Clapeyron amplification (Fernández-Alvarez et al., 2023). From the terrestrial part, soil moisture climatology will generally show trends with future climate change (Lai et al., 2023), meaning that the absolute and relative contributions from ocean to precipitation should increase (for drying trends) or decrease (for wetting trends) with global warming. Deviations from a soil-moisture based null model may point at the role of regional terrestrial conditions such as land cover changes. Also, moisture tracking forced by stylized climate- and land cover change experiments in Earth System Models would be a significant step forward in this regard.

### 4.2 Regional differences

In contrast to the globally consistent pattern of decreasing precipitation recycling ratios with more severe SSPs, there are large spatial differences. These spatial differences are broadly consistent among SSPs, although much more pronounced in the most severe scenarios SSP3-7.0 and SSP5-8.5. Overall, independent of the scenario, we find that regional drying tends to be dominated by reduced recycling from land, and regional wetting is dominated by an increase in moisture from the ocean. We call drying land-dominated if it coincides with a significant increase in terrestrial precipitation recycling ratio and we call it ocean-dominated if it coincides with a significant decrease in terrestrial precipitation recycling ratio. Similarly, wetting can be land- or ocean-dominated. The increasing moisture flux from the ocean towards land (Wang et al., 2023) leads to increasing

precipitation especially in the higher northern latitudes and in south and east Asia, where precipitation recycling tends to increase in an absolute sense, but not always in a relative sense. For instance, looking at the major south and east Asian major river basins, terrestrial precipitation recycling ratios will tend to go up for the Amur and Indus basins, but down for the Yangtze, Huang He (Yellow River), and Ganges basins. A notable example for the temperate boreal zone where precipitation recycling may decrease in a significant way is eastern Europe. This is reflected by the major decrease in terrestrial precipitation recycling ratio from 38% to 33% for the Danube basin in SSP5-8.5 by the 2090s.

The Amazon is projected to face severe declines in land-derived precipitation, except in SSP1-2.6. Even in SSP2-4.5, the southern Amazon will receive up to 300 mm year$^{-1}$ less precipitation from land. With more severe climate change, the area facing similar declines expands westward and northward. Given the relatively high recycling within the western Amazon related to the presence of the Andes (Staal et al., 2018), the high deforestation rates in SSP3-7.0 would explain some of the recycling changes there (Li et al., 2023b). For the Amazon as a whole, the effects of deforestation on moisture recycling seem to be overshadowed by those of increasing atmospheric $CO_2$ concentrations, although in terms of recycling ratios, the larger deforestation in SSP3-7.0 compensates for the stronger radiative forcing in SSP5-8.5. This is consistent with results from CMIP5 models, where every 10% of the basin deforested leads to an average precipitation decline of only 1.6% (Spracklen and Garcia-Carreras, 2015), and where severe climate change (RCP8.5) leads to increased moisture influx from the Atlantic into South America, but reduced precipitation (recycling) over the Amazon (Arias et al., 2023). The comparatively small effect of deforestation is further confirmed by experiments using a range of CMIP6 models, where one study found that warming, via atmospheric circulation changes, accounts for 55% of Amazon drying under SSP3-7.0 conditions. The remaining 45% mostly resulted from the physiological effects of $CO_2$ increase rather than deforestation directly (Li et al., 2023b; see also Skinner et al., 2017). A regional atmospheric model also indicates that physiological effects on Amazonian transpiration and precipitation of a factor 1.5 increase of atmospheric $CO_2$ would be similar to those of 100% deforestation (Sampaio et al., 2021). Another study of CMIP6 model outputs reports a multi-model average reduction of 17.5% in terrestrial moisture recycling ratio in SSP5-8.5 by the end of the century (Baker and Spracklen, 2022), which is considerably larger than our estimate of 7%. It must be noted though, that the range among Earth System Models of projections of Amazon precipitation response to deforestation in the Amazon is very large (Luo et al., 2022) and that these models tend to project a linear response (Spracklen and Garcia-Carreras, 2015). This linearity contrasts strongly with inferred strongly nonlinear responses in different types of models (Baudena et al., 2021; Bochow and Boers, 2023), making it very plausible that land-cover change effects on Amazon precipitation recycling are underestimated in our study.

Dominance of atmospheric $CO_2$ increase over regional land-cover change is also inferred for the Congo, albeit this dominance is manifested differently than in the Amazon. The Congo basin is a hotspot of deforestation in the more severe scenarios SSP3-7.0 and SSP5-8.5. In particular, deforestation of more than half of the current forest cover in the Congo basin is projected for SSP3-7.0 by the end of the century, which is expected to have large effects on precipitation (Luo et al., 2022; Smith et al., 2023). Indeed, the large-scale land conversion in this scenario coincides with a drop in basin precipitation recycling ratio from 33% to 30%. However, this is compensated by an increased influx of moisture from the Atlantic Ocean (Baker and Spracklen, 2022). Contrasting with recent drying in some part of the basin (Vizy et al., 2023), in CMIP6 models including in NorESM2, increasing precipitation levels are projected for the Congo (Baker and Spracklen, 2022; Staal et al., 2020b), although our estimate of a 7% reduction in terrestrial precipitation recycling ratio is lower than the CMIP6 average of 12% (Baker and Spracklen, 2022). Despite overall projections of precipitation increases, in the hypothetical case that large-scale deforestation would occur but with comparatively little additional $CO_2$ emissions, precipitation levels in the Congo could decrease significantly (Staal et al., 2020b).

For most areas across the globe, the sign of change in absolute precipitation recycling coincides with the sign of change in precipitation itself. Despite this agreement in an absolute sense, whether drying or wetting is land-dominated or ocean-dominated does differ across the globe. Land dominance may happen in regions where land-cover changes are so severe that they greatly affect evapotranspiration rates. If the influx of moisture from the oceans does not change, then a decrease or increase of continental evapotranspiration would result in land dominance of drying or wetting. A change in oceanic influx can similarly be expected to result in ocean dominance of either drying or wetting. Increasing length scales of moisture recycling resulting from elevated residence times in a warmer atmosphere are also expected to extend the oceanic influence further into the continents, but no clear signal of this can be seen in the vicinity of the coastlines. Land dominance, visible most clearly for SSP5-8.5, tends to occur in regions with already large terrestrial precipitation recycling ratios, mainly interior South America (land-dominated drying) and eastern Asia (land-dominated wetting). Land-dominated drying may also happen in eastern Europe (although not under the two mildest scenarios), in Central America, and in subtropical Sub-Saharan Africa. Ocean-dominance, mainly in the form of wetting, is found primarily in the high northern latitudes and in central Africa, the latter of which is in line with projected increasing moisture influx and deforestation. Globally, the patterns of wetting and drying are consistent with CMIP6 averages (Cook et al., 2020).

Regional precipitation variations have both social and ecological implications. Among the social implications are variations in local and remote water provision; among the ecological implications are those on forest and biodiversity intactness. Some regions that are highly dependent on rainfed agriculture, such as Sub-Saharan Africa, might be heavily impacted by changes in terrestrial moisture recycling (Nyasulu et al., 2024). However, even though changes in terrestrial precipitation recycling seem to be strongly impacted by global warming, we can use insights on the changing moisture recycling patterns to influence precipitation levels where they are most needed. With studies like these, we are gradually becoming better able to understand how deforestation, but importantly also potential reforestation, may influence precipitation patterns regionally. Thus, we could incorporate precipitation enhancements in strategic decisions of restoring global forest land (Staal et al., 2024).

### 4.3 Limitations

Our baseline estimate of 34% global precipitation recycling is lower than published estimates: the one by Van der Ent et al. (2010) of 40% is comparable to ours, but more often precipitation recycling is estimated to be higher, ranging from 51% (Tuinenburg et al., 2020) and 54.5% (Gimeno et al., 2020) to 62% (Cheng and Lu, 2023). Our baseline estimate of 63% for global evaporation recycling is in between that of 57% in Van der Ent et al. (2010) and 70% in Tuinenburg et al. (2020). The ratio of our baseline evapotranspiration over precipitation (ratios) is 0.52. Compared to 0.70 (Van der Ent et al., 2010) and 0.73 (Tuinenburg et al., 2020) this is indeed low, although Oki & Kanae (2006) report a more comparable ratio of 0.59. These low values in our results may be due to overestimations of precipitation over ocean in NorESM2, leading to larger recycling of oceanic water vapor and reduced ocean-to-land moisture transport (Seland et al., 2020).

Regardless of uncertainties that surround evapotranspiration values, we can evaluate their trends. Globally, evapotranspiration has been increasing over the recent decades and is expected to keep increasing in all future scenarios (Yang et al., 2023). Also in our data, evapotranspiration is projected to increase except in SSP5-8.5. The main cause of projected evapotranspiration increases is global greening (Yang et al., 2023), a consequence of $CO_2$ fertilization at the global scale and of human activities at regional scales (Piao et al., 2019; Zhu et al., 2016). Around one-fifth of global precipitation has been evaporated directly by vegetation (Keys et al., 2016). Global greening, measured as an increase in Leaf Area Index, has already stimulated terrestrial moisture recycling since at least the beginning of this century, compensating for the drying effects of deforestation in various regions such as the Amazon (Cui et al., 2022). Although this greening trend is expected to continue into the future, the effect of greening on evapotranspiration, and consequently on moisture recycling, may decline as more $CO_2$ builds up in the

atmosphere (particularly in case of SSP5-8.5), which reduces leaf stomatal conductance (Yang et al., 2023). Meanwhile, however, simultaneous increases in vapor pressure deficit may maintain evapotranspiration levels despite $CO_2$ fertilization (Li et al., 2023a). Leaf processes in the land model underlying NorESM2, CLM5, are based on the Medlyn stomatal conductance model (Medlyn et al., 2011). The CLM5 calibration emphasizes historical and transient effects of $CO_2$- and N-fertilization on evapotranspiration via stomatal conductance, and improvements are needed to capture broader biogeochemical feedback mechanisms (Fisher et al., 2019; Franks et al., 2017).

We analyzed mean annual (and mean seasonal) precipitation recycling without accounting for interannual variability. The main mode of interannual climatic variability is the El Niño Southern Oscillation (ENSO). During El Niño phases, weather patterns across large parts of the globe are disrupted, and these disruptions are expected to intensify this century (Power et al., 2013). During El Niño years of the past decades, moisture recycling over the northern hemisphere and in the tropics tended to be reduced, while that in the southern hemisphere tended to be enhanced (Posada-Marín et al., 2023). The sign of the anomaly of precipitation recycling mostly agrees with the sign of the precipitation anomaly itself, and the relative contributions of moisture recycling reductions in the northern hemisphere were found to be larger than the relative contributions of moisture recycling increases in the southern hemisphere (Posada-Marín et al., 2023). If this pattern continues in the future, then increasingly strong El Niño phases can be expected to result in increasingly strong but opposite moisture recycling anomalies in both hemispheres. NorESM2 does capture ENSO events well compared to other CMIP6 models, but it also overestimates sea surface temperatures during these phases (Seland et al., 2020) and poorly captures the relationship between ENSO and the South Pacific Quadrupole, which is a key driver for onset of ENSO events (Wang et al., 2021). Furthermore, the model generates internal climate variability. Three ensemble members of historical runs gave around 0.1 °C difference in global temperature on a yearly basis (Seland et al., 2020). Such ensemble runs allow for the separation of internal climate variability from the forced climatic signal. In the future, different ensemble members can be used to force UTrack, so the signal and noise in precipitation recycling can be separated, even though it is unlikely that the global trends and patterns would be affected by different ensemble members.

We used the output of only one Earth System Model to force the UTrack moisture tracking model, since it was the only one providing the required variables at high temporal and spatial resolution for the Tier 1 scenarios. Projections of future climate change, including the hydrological cycle, are notoriously different among Earth System Models (Wu et al., 2024). Any bias in NorESM2 or its underlying components that affects the atmosphere or land-atmosphere interactions may propagate to our results. Similar studies would be needed using different models to better understand the uncertainties in future changes in terrestrial moisture recycling. The most similar study to ours is by Findell et al. (2019), who used the GFDL-ESM2G Earth System Model to study global moisture recycling. However, they did not differentiate among different SSPs. Baker & Spracklen (2022) and Arias et al. (2023) used Earth System Models to study changes in moisture recycling for particular regions.

**5 Conclusions**

We studied how terrestrial moisture recycling may develop globally towards the end of the 21$^{st}$ century. We developed a new version of the Lagrangian atmospheric moisture tracking model UTrack, forced by output of the Norwegian Earth System Model version 2. We performed forward tracking of evaporation from the global land area and from the 26 major river basins of the world in four combined climate-change and land-cover change scenarios: SSP1-2.6, SSP2-4.5, SSP3-7.0, and SSP5-8.5. We find that, globally, terrestrial moisture recycling decreases by 1.5% with every degree of warming. In the most severe scenarios (SSP3-7.0 and SSP5-8.5), regional drying accompanied by reductions of moisture supply over land are projected

for, for instance, the Amazon and eastern Europe. Especially in high norther latitudes, wetting is projected in these scenarios, sometimes accompanied by increases of moisture from land and sometimes from ocean. Although both land cover changes and global climate change affect moisture recycling over land areas, most of the differences in moisture recycling among the various SSPs over the 21$^{st}$ century seem to be caused by direct and indirect effects of global warming.

## Appendix A: Supplementary tables and figures

**Table A1.** Basin and terrestrial precipitation recycling ratios (%) for the 26 major river basins of the world in the baseline scenario (SSP2-4.5 during 2015–2024) and by the middle of the century (2050–2059) in SSP1-2.6, SSP2-4.5. SSP3-7.0, and SSP5-8.5. Significant differences with the baseline are indicated by one (for $p < 0.05$) or two (for $p < 0.01$) asterisks.

|  | Baseline | | SSP1-2.6 | | SSP2-4.5 | | SSP3-7.0 | | SSP5-8.5 | |
|---|---|---|---|---|---|---|---|---|---|---|
|  | Basin | Terr. | Basin | Terr. | Basin | Terr. | Basin | Terr. | Basin | Terr. |
| Amazon | 27 | 43 | 26 | 43 | 26 | 43 | 25* | 42 | 25** | 42 |
| Amur | 19 | 65 | 20** | 71 | 19 | 68 | 18 | 68 | 18 | 66 |
| Chad | 16 | 59 | 14* | 57 | 14* | 57 | 15* | 56 | 14* | 56 |
| Congo | 33 | 56 | 33 | 57 | 32 | 55 | 32* | 53 | 33 | 55 |
| Danube | 9 | 38 | 9 | 37 | 10 | 40 | 8* | 33** | 10 | 38 |
| Euphrates-Tigris | 8 | 35 | 8 | 39 | 9 | 37 | 9 | 39 | 8 | 41 |
| Ganges | 9 | 46 | 9 | 44 | 9 | 45 | 9 | 44 | 8 | 43 |
| Huang He | 19 | 82 | 18 | 82 | 17** | 80 | 17* | 79 | 17** | 82 |
| Indus | 16 | 73 | 17 | 85 | 18 | 87* | 16 | 81 | 17 | 89 |
| Kolyma | 5 | 33 | 5 | 35* | 5 | 35 | 5 | 34 | 4 | 34 |
| Lena | 16 | 59 | 16 | 60 | 15 | 58 | 15 | 59 | 14** | 56 |
| Mackenzie | 16 | 39 | 15 | 37 | 15 | 37 | 15 | 36 | 13** | 34** |
| Mississippi | 15 | 42 | 15 | 41 | 15 | 40* | 15 | 40 | 14 | 39* |
| Murray | 9 | 28 | 9 | 30 | 9 | 29 | 9 | 29 | 9 | 30 |
| Nelson | 10 | 50 | 10 | 47 | 9* | 46* | 9* | 46** | 9** | 45* |
| Niger | 11 | 45 | 11 | 45 | 10** | 44 | 11** | 44 | 10* | 42 |
| Nile | 24 | 57 | 23 | 59 | 23 | 56 | 23 | 56 | 24 | 59 |
| Ob | 11 | 45 | 12 | 47 | 11 | 46 | 11 | 44 | 11 | 45 |
| Orange | 16 | 62 | 15 | 59 | 14 | 59 | 15 | 61 | 15 | 60 |
| Paraná | 24 | 66 | 24 | 66 | 23 | 65 | 23 | 66 | 23 | 65 |
| Saint Lawrence | 8 | 43 | 9 | 44 | 8 | 40* | 8* | 41 | 8 | 40 |
| Volga | 8 | 39 | 9 | 41 | 8 | 38 | 8 | 39 | 8 | 38 |
| Yangtze | 19 | 75 | 19 | 73 | 19 | 75 | 18 | 73 | 18 | 72 |
| Yenisey | 14 | 57 | 14 | 59** | 13* | 56 | 13 | 56 | 13 | 56 |
| Yukon | 8 | 27 | 7 | 26 | 7 | 26 | 6* | 25 | 7 | 26 |
| Zambezi | 15 | 55 | 16 | 53* | 15* | 52* | 15** | 52 | 15* | 53 |

**Table A2.** Forest and cropland cover (%) of the global non-Antarctic land surface and the 26 major river basins of the world in the baseline scenario (SSP2-4.5 during 2015–2024) and by the middle of the century (2050–2059) in SSP1-2.6, SSP2-4.5. SSP3-7.0, and SSP5-8.5.

| | Baseline | | SSP1-2.6 | | SSP2-4.5 | | SSP3-7.0 | | SSP5-8.5 | |
|---|---|---|---|---|---|---|---|---|---|---|
| | Forest | Crop | Forest | Crop | Forest | Crop | Forest | Crop | Forest | Crop |
| Global land | 25 | 11 | 27 | 10 | 25 | 12 | 24 | 12 | 24 | 12 |
| Amazon | 82 | 3 | 82 | 3 | 80 | 4 | 79 | 6 | 82 | 3 |
| Amur | 43 | 10 | 46 | 9 | 44 | 8 | 44 | 7 | 43 | 10 |
| Chad | 4 | 10 | 5 | 10 | 4 | 13 | 4 | 11 | 4 | 11 |
| Congo | 60 | 4 | 60 | 5 | 59 | 6 | 49 | 6 | 49 | 7 |
| Danube | 31 | 34 | 38 | 31 | 30 | 35 | 35 | 27 | 31 | 34 |
| Euphrates-Tigris | 0 | 17 | 0 | 19 | 0 | 16 | 0 | 21 | 0 | 12 |
| Ganges | 15 | 51 | 16 | 49 | 15 | 51 | 13 | 56 | 13 | 60 |
| Huang He | 7 | 15 | 12 | 12 | 6 | 15 | 8 | 13 | 7 | 16 |
| Indus | 5 | 28 | 6 | 27 | 5 | 29 | 5 | 32 | 5 | 30 |
| Kolyma | 9 | 0 | 9 | 0 | 9 | 0 | 9 | 0 | 9 | 0 |
| Lena | 54 | 0 | 54 | 0 | 54 | 0 | 54 | 0 | 54 | 0 |
| Mackenzie | 46 | 2 | 46 | 2 | 46 | 3 | 46 | 2 | 46 | 2 |
| Mississippi | 17 | 30 | 23 | 27 | 20 | 30 | 15 | 30 | 16 | 29 |
| Murray | 13 | 21 | 14 | 21 | 12 | 22 | 13 | 21 | 13 | 19 |
| Nelson | 31 | 36 | 37 | 32 | 31 | 37 | 32 | 34 | 32 | 35 |
| Niger | 4 | 25 | 6 | 25 | 3 | 31 | 2 | 38 | 2 | 29 |
| Nile | 6 | 16 | 7 | 17 | 6 | 19 | 4 | 18 | 5 | 21 |
| Ob | 33 | 15 | 35 | 15 | 34 | 12 | 33 | 12 | 33 | 15 |
| Orange | 1 | 5 | 2 | 5 | 1 | 7 | 1 | 20 | 1 | 8 |
| Paraná | 30 | 23 | 32 | 22 | 27 | 28 | 29 | 24 | 27 | 32 |
| Saint Lawrence | 44 | 12 | 46 | 11 | 45 | 12 | 42 | 14 | 44 | 12 |
| Volga | 44 | 25 | 45 | 26 | 44 | 24 | 46 | 22 | 44 | 24 |
| Yangtze | 40 | 16 | 49 | 16 | 40 | 18 | 50 | 11 | 40 | 20 |
| Yenisey | 55 | 2 | 55 | 2 | 54 | 3 | 55 | 1 | 55 | 2 |
| Yukon | 36 | 0 | 36 | 0 | 36 | 0 | 36 | 0 | 36 | 0 |
| Zambezi | 25 | 9 | 35 | 10 | 25 | 10 | 17 | 23 | 24 | 13 |

**Table A3.** Forest and cropland cover (%) of the global non-Antarctic land surface and the 26 major river basins of the world in the baseline scenario (SSP2-4.5 during 2015–2024) and by the end of the century (2090–2099) in SSP1-2.6, SSP2-4.5. SSP3-7.0, and SSP5-8.5.

| | Baseline | | SSP1-2.6 | | SSP2-4.5 | | SSP3-7.0 | | SSP5-8.5 | |
|---|---|---|---|---|---|---|---|---|---|---|
| | Forest | Crop | Forest | Crop | Forest | Crop | Forest | Crop | Forest | Crop |
| Global land | 25 | 11 | 27 | 12 | 25 | 13 | 23 | 14 | 25 | 12 |
| Amazon | 82 | 3 | 82 | 4 | 82 | 5 | 75 | 11 | 82 | 3 |
| Amur | 43 | 10 | 47 | 9 | 45 | 7 | 45 | 7 | 43 | 10 |
| Chad | 4 | 10 | 6 | 10 | 4 | 14 | 3 | 15 | 4 | 11 |
| Congo | 60 | 4 | 61 | 7 | 59 | 9 | 27 | 8 | 50 | 8 |
| Danube | 31 | 34 | 38 | 35 | 36 | 29 | 37 | 22 | 31 | 34 |
| Euphrates-Tigris | 0 | 17 | 0 | 19 | 0 | 16 | 0 | 22 | 0 | 10 |
| Ganges | 15 | 51 | 17 | 49 | 16 | 51 | 12 | 58 | 13 | 60 |
| Huang He | 7 | 15 | 14 | 11 | 7 | 15 | 9 | 12 | 7 | 16 |
| Indus | 5 | 28 | 6 | 27 | 5 | 29 | 5 | 34 | 5 | 30 |
| Kolyma | 9 | 0 | 9 | 0 | 9 | 0 | 9 | 0 | 9 | 0 |
| Lena | 54 | 0 | 54 | 0 | 54 | 1 | 54 | 0 | 54 | 0 |
| Mackenzie | 46 | 2 | 46 | 2 | 46 | 3 | 44 | 4 | 46 | 2 |
| Mississippi | 17 | 30 | 22 | 31 | 23 | 31 | 15 | 30 | 16 | 29 |
| Murray | 13 | 21 | 17 | 20 | 12 | 25 | 13 | 20 | 13 | 18 |
| Nelson | 31 | 36 | 40 | 29 | 32 | 36 | 32 | 31 | 32 | 35 |
| Niger | 4 | 25 | 7 | 29 | 2 | 34 | 1 | 57 | 2 | 30 |
| Nile | 6 | 16 | 10 | 18 | 7 | 22 | 4 | 20 | 9 | 21 |
| Ob | 33 | 15 | 35 | 15 | 34 | 12 | 33 | 13 | 33 | 14 |
| Orange | 1 | 5 | 2 | 6 | 1 | 10 | 1 | 24 | 1 | 8 |
| Paraná | 30 | 23 | 32 | 27 | 29 | 34 | 29 | 25 | 28 | 32 |
| Saint Lawrence | 44 | 12 | 45 | 12 | 46 | 12 | 42 | 15 | 43 | 12 |
| Volga | 44 | 25 | 45 | 30 | 46 | 24 | 47 | 22 | 44 | 24 |
| Yangtze | 40 | 16 | 50 | 20 | 42 | 17 | 53 | 7 | 40 | 20 |
| Yenisey | 55 | 2 | 55 | 2 | 55 | 4 | 55 | 1 | 55 | 2 |
| Yukon | 36 | 0 | 36 | 0 | 35 | 1 | 36 | 0 | 36 | 0 |
| Zambezi | 25 | 9 | 39 | 10 | 25 | 15 | 16 | 31 | 30 | 13 |

**Table A4.** The percentage of global terrestrial grid cells with a significant change in annual precipitation, its direction of change (drying or wetting), and its direction of change in terrestrial precipitation recycling (TPR) ratio by the end of the century (2090–2099) in SSP1-2.6, SSP2-4.5. SSP3-7.0, and SSP5-8.5 compared to the baseline scenario (SSP2-4.5 during 2015–2024).

| | SSP1-2.6 | SSP2-4.5 | SSP3-7.0 | SSP5-8.5 |
|---|---|---|---|---|
| Significant change in precipitation | 8.7% | 17.1% | 30.0% | 41.5% |
| of which drying | 27.2% | 22.7% | 16.4% | 19.0% |
| of which increasing TPR ratio | 56.6% | 55.7% | 30.0% | 24.5% |
| of which decreasing TPR ratio | 45.4% | 44.3% | 70.0% | 75.5% |
| of which wetting | 72.8% | 77.3% | 83.6% | 81.0% |
| of which increasing TPR ratio | 62.1% | 50.1% | 43.5% | 32.9% |
| of which decreasing TPR ratio | 37.9% | 49.9% | 56.5% | 67.1% |

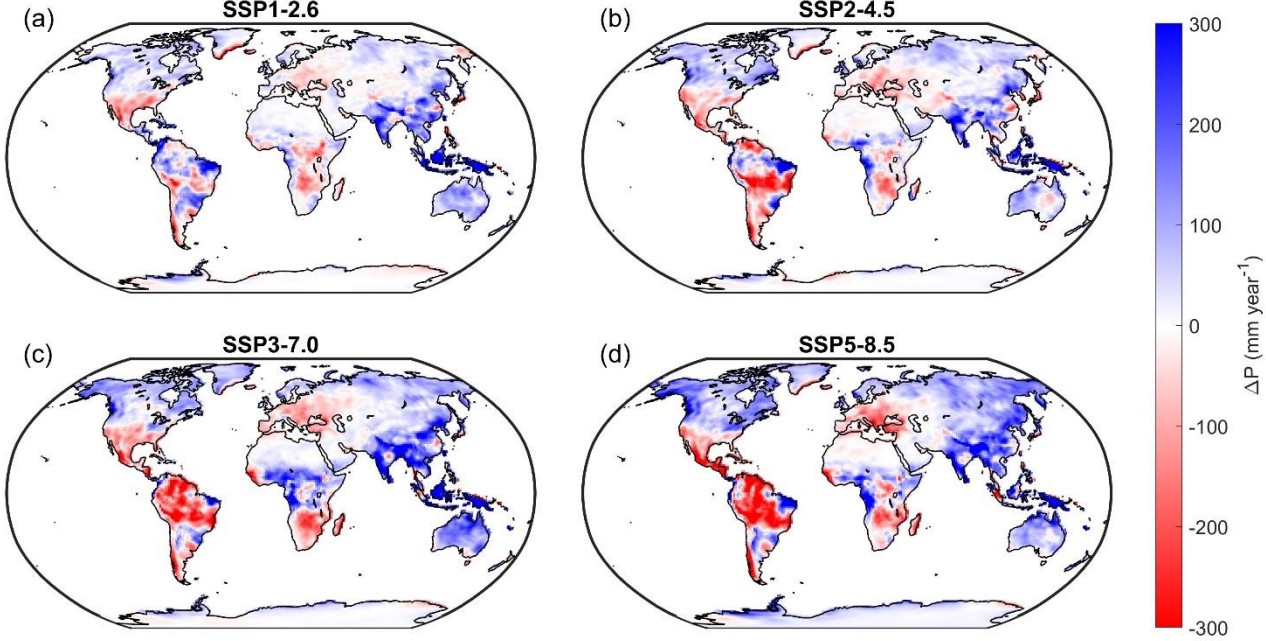

**Figure A1.** Differences in annual precipitation (ΔP) across the globe between the baseline period (2015–2024) and the end of the century (2090–2099) in mm year[-1], for (a) SSP1-2.6, (b) SSP2-4.5, (c) SSP3-7.0, and (d) SSP5-8.5. Color scales are truncated at -300 mm year[-1] and 300 mm year[-1].

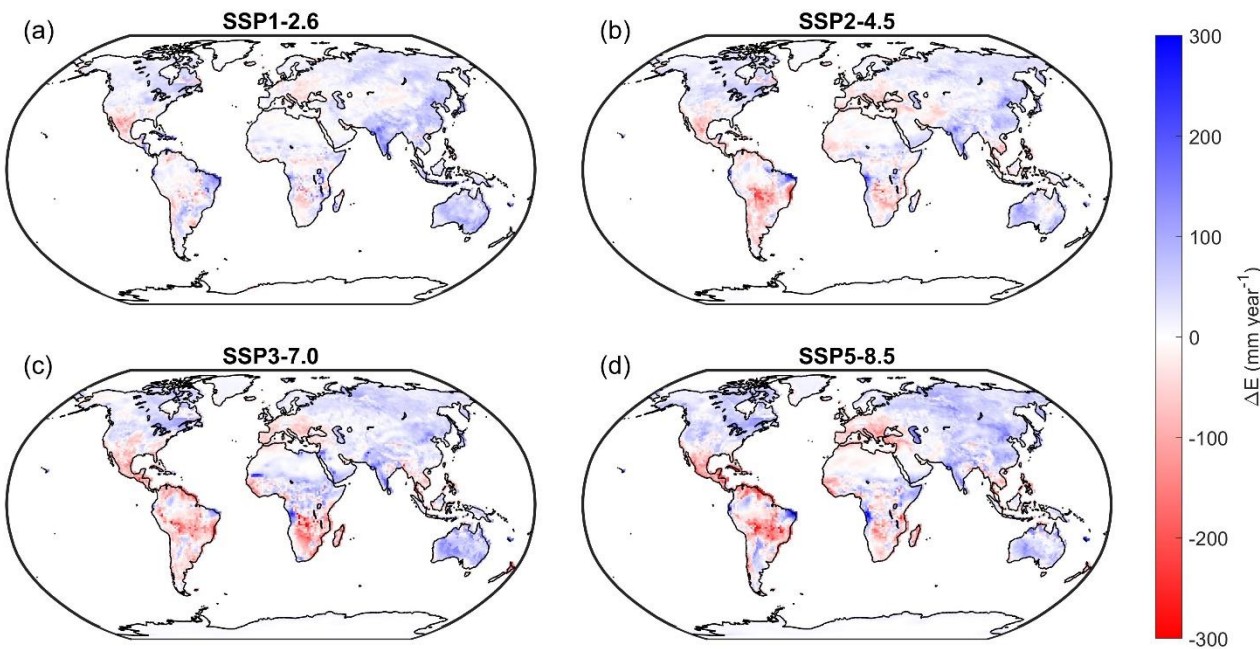


**Figure A2.** Differences in annual evaporation (ΔE) across the globe between the baseline period (2015–2024) and the end of the century (2090–2099) in mm year[-1], for (a) SSP1-2.6, (b) SSP2-4.5, (c) SSP3-7.0, and (d) SSP5-8.5. Color scales are truncated at -300 mm year[-1] and 300 mm year[-1].

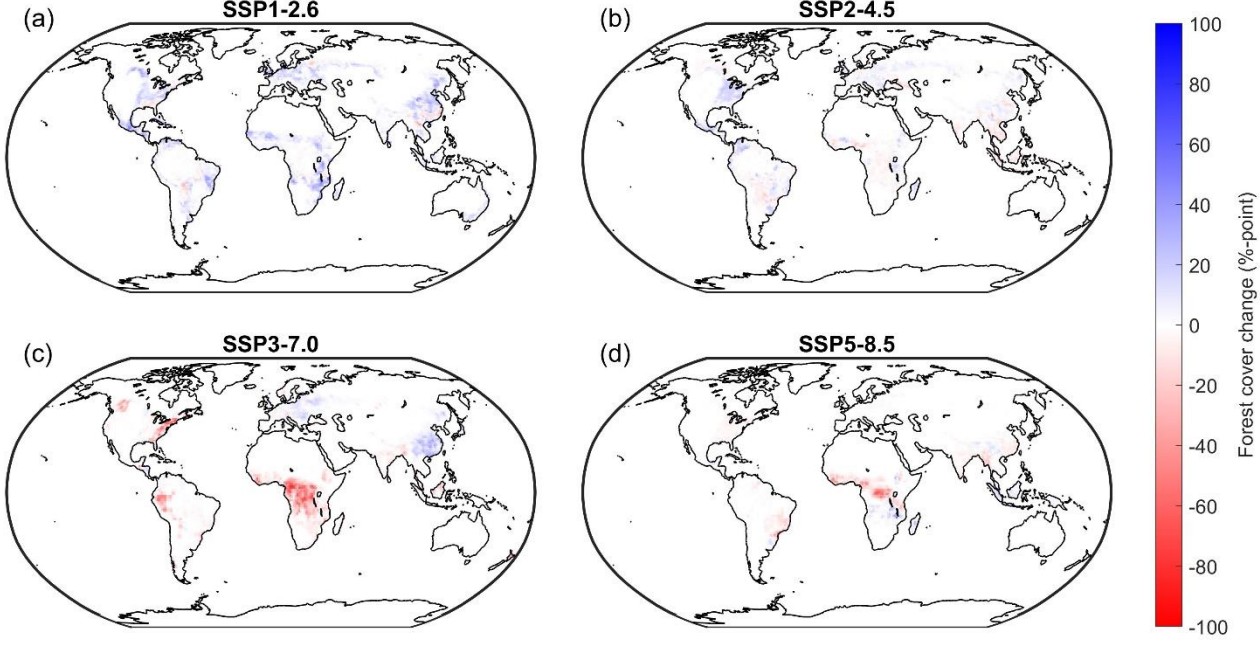

**Figure A3.** Differences in forest cover (%-point) between the baseline period (2015–2024) and the end of the century (2090–2099) for (a) SSP1-2.6, (b) SSP2-4.5, (c) SSP3-7.0, and (d) SSP5-8.5.

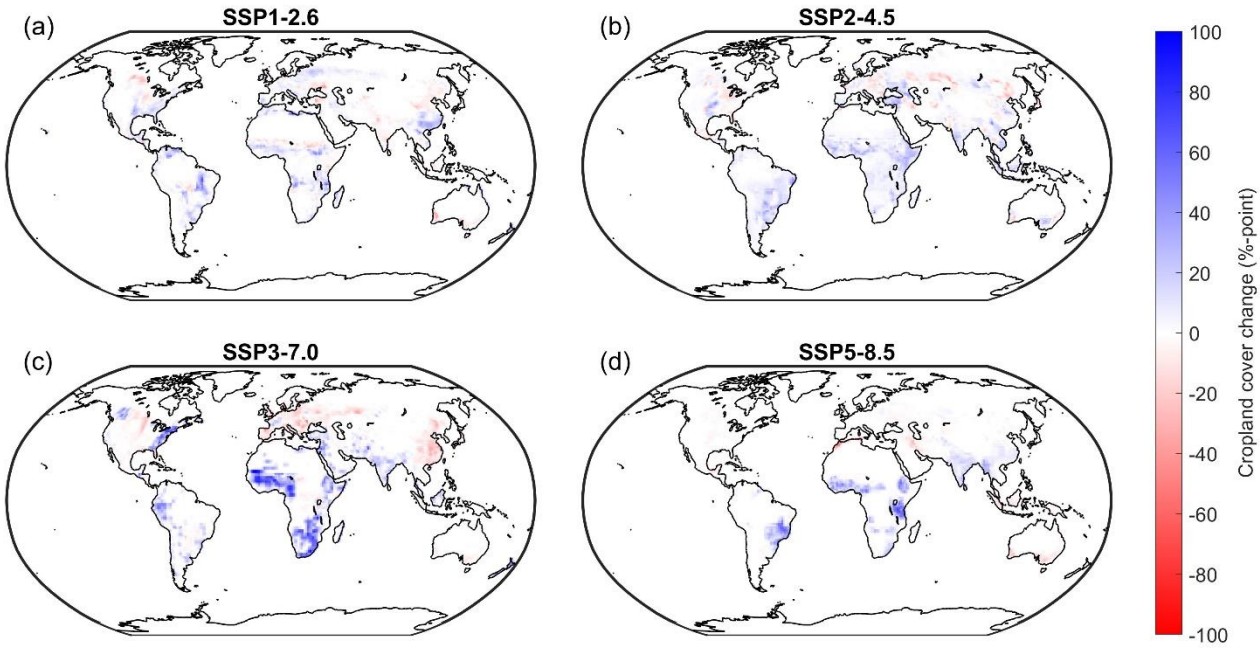

**Figure A4.** Differences in cropland cover (%-point) between the baseline period (2015–2024) and the end of the century (2090–2099) for (a) SSP1-2.6, (b) SSP2-4.5, (c) SSP3-7.0, and (d) SSP5-8.5.

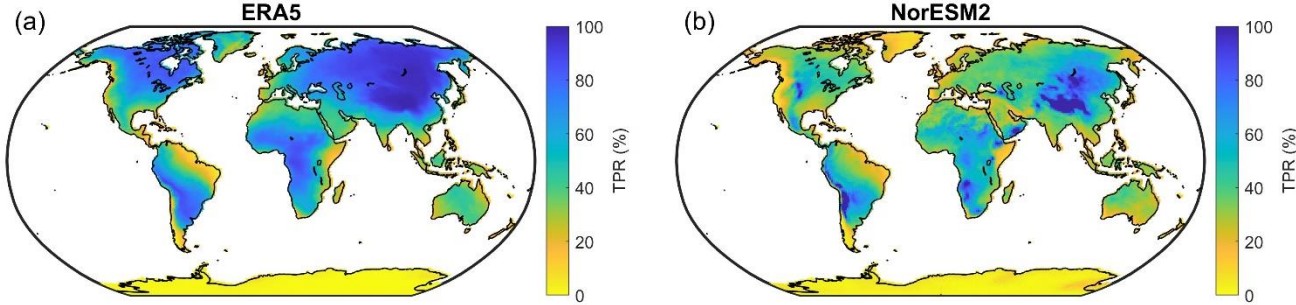

**Figure A5.** Terrestrial precipitation recycling ratios according to UTrack using ERA5 input and NorESM2 input. (a) Terrestrial moisture recycling ratio (%) from Tuinenburg et al. (2020), based on ERA5 data from 2008-2017. (b) Terrestrial moisture recycling ratio (%) in the SSP2-4.5 scenario for 2015-2024.


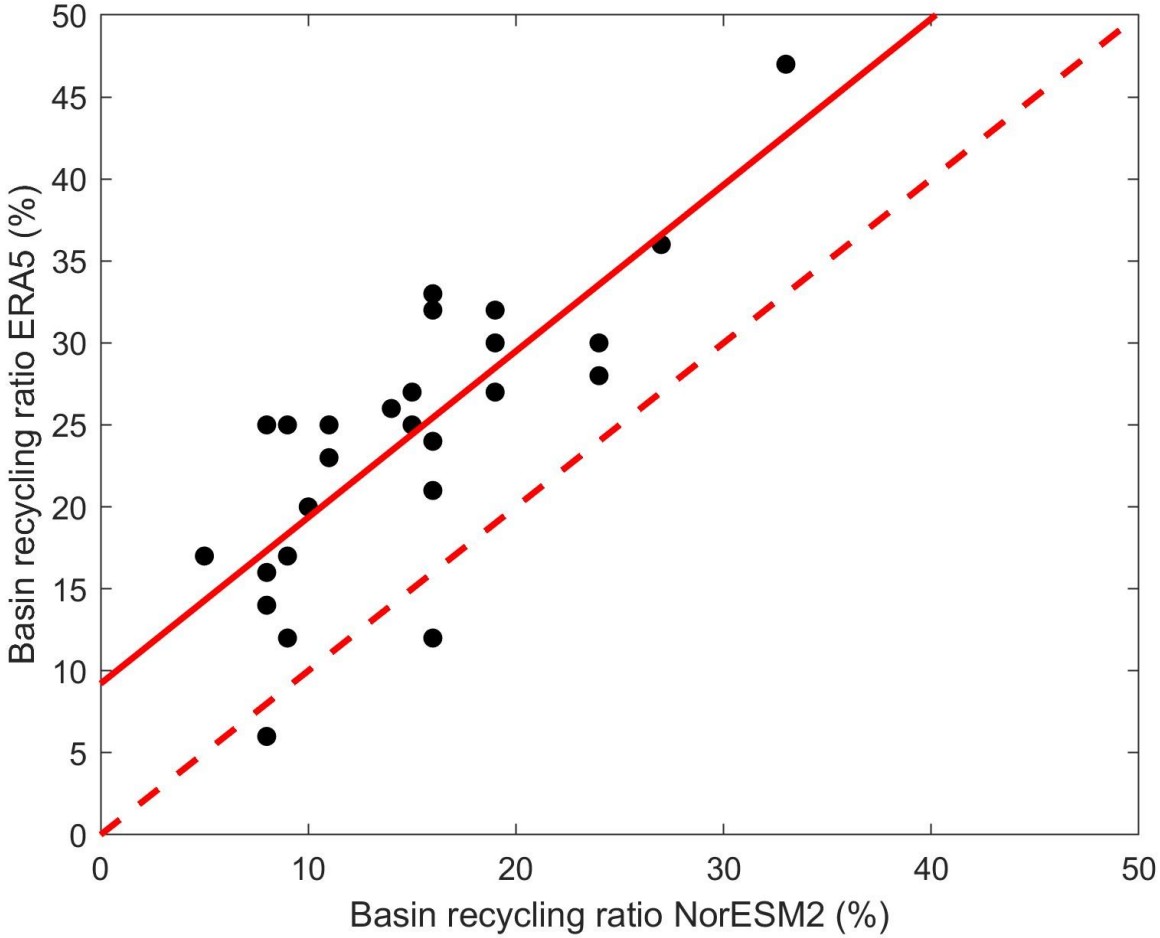

**Figure A6.** Basin precipitation recycling ratios (%) for the 26 major river basins of the world (also see Table 2) according to UTrack using ERA5 input and NorESM2 input. A linear regression (red solid line) gives $R^2 = 0.62$. The y=x line gives $R^2 = 0.38$. On average, the ERA5-based basin recycling ratios are 9.4 percentage point larger than those based on NorESM2.


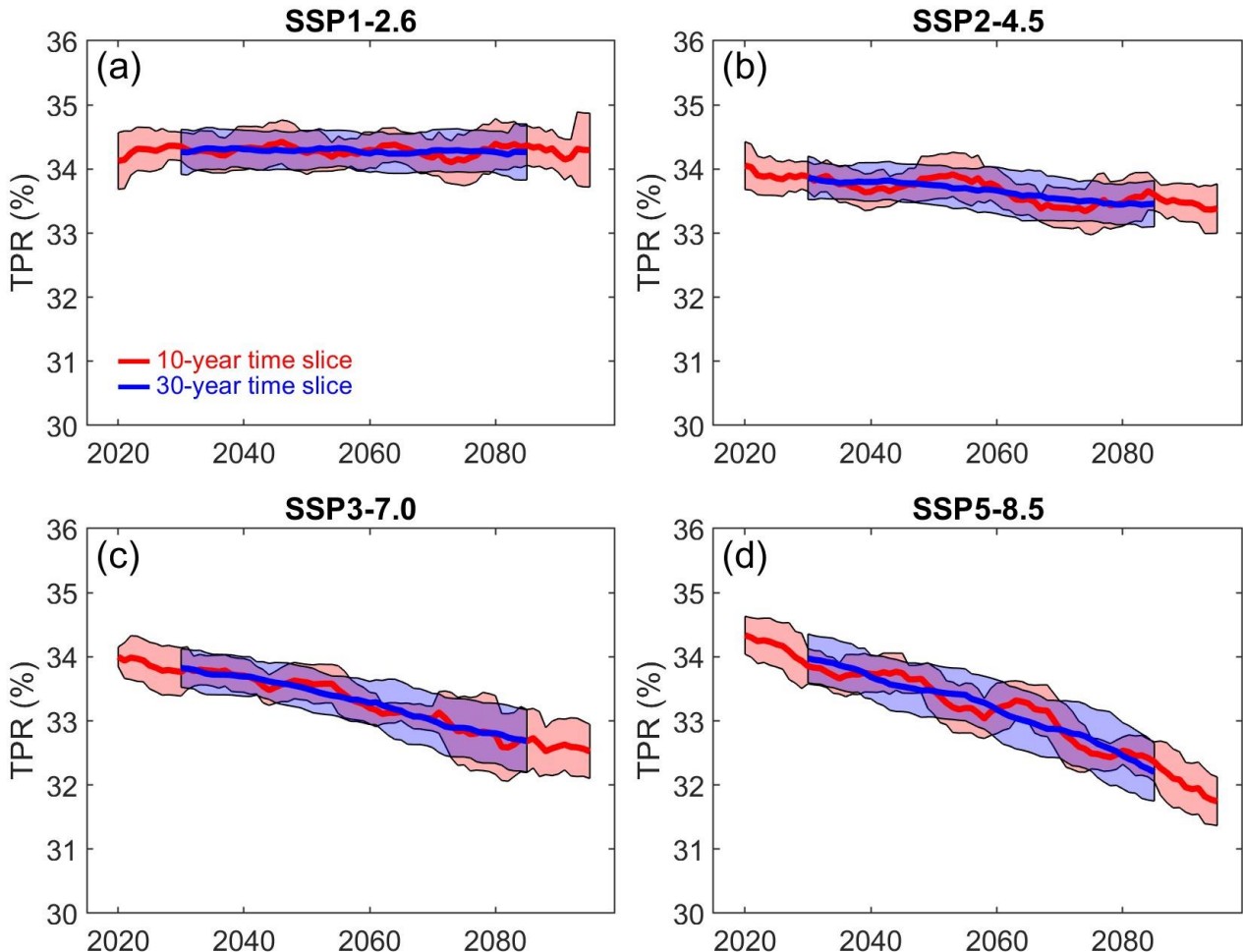

**Figure A7.** Moving averages of the global terrestrial precipitation recycling ratio (TPR, in %) ± one standard deviation for ten-year time slices (in red) and 30-year time slices (in blue), for (a) SSP1-2.6, (b) SSP2-4.5, (c) SSP3-7.0, and (d) SSP5-8.5. Results for 2015–2099 were used, meaning that the moving averages for ten-year time slices range between 2020–2095 and those for 30-year time slices between 2030–2085.

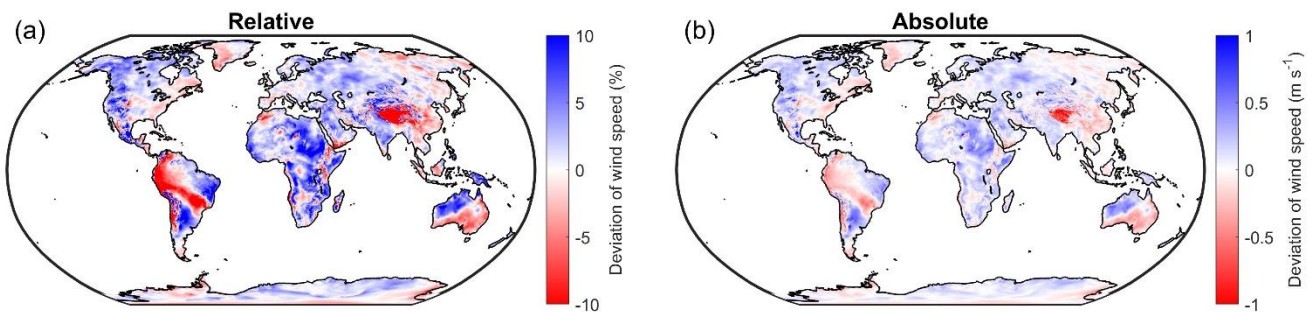

**Figure A8.** Deviations of instantaneous wind speed at 00Z with average hourly wind speed across the day in ERA5 based on the period 2010–2023. a) Relative deviations (%), b) Absolute deviations (m/s).

## Code availability

The model code used for this study is available at the following URL: https://github.com/ArieStaal/UTrack-NorESM2_global_land.

## Data availability

The monthly data for global precipitation recycling and basin recycling for the 26 major river basins of the world that were used in this study are available for download at the following URL: https://zenodo.org/records/10650579.

## Author contributions

AS and OAT conceived the study. AS and SCD designed the study. PM, AS, and OAT developed the model code. AS carried out the simulations. AS and MKN carried out the analyses. AS wrote the paper with contributions from all authors.

## Competing interests


Some authors are members of the editorial board of *Earth System Dynamics*.

## Acknowledgements

AS acknowledges funding from the Dutch Research Council (NWO) Talent Program Grant VI.Veni.202.170. MKN acknowledges funding from Swedish Research Council for Sustainable Development (FORMAS) project number 2017-
675 01033.

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
