# Peer review of "Global terrestrial moisture recycling in Shared Socioeconomic Pathways"

_EGUsphere, 2024_

## Author Comment (AC1)

**Summary**

The authors use future projections from the NorESM2 climate model to drive the atmospheric moisture tracking model UTrack. The future projections represent four possible socioeconomic scenarios of varying severity, with the aim of estimating global responses in terrestrial precipitation recycling to different emissions pathways. The authors present their new version of the UTrack model in a Github repository. The paper is well written and a valuable addition to the literature. The introduction and discussion sections were especially good, however, the description of the results could be improved to help the reader make links between the text and the figures more readily, and to ensure your findings come across in the clearest possible way.

We thank the reviewer for their kind words and the constructive and detailed comments. They will help us improve the manuscript, in particular te clarity and readability of the Results. Below we respond briefly to the individual comments and how we propose to deal with them.

Line by line comments

Line 74 – add some examples of studies that use moisture tracking models to assess precipitation recycling

We will add several examples, from local to global examples, and with different purposes such as understanding the role of ecosystem services.

Line 97 – can you add further explanation for why 100 parcels are released at random locations above the starting area? For evaporation, shouldn't parcels come from the surface, and precipitation higher up in the atmosphere? You say further up in the paragraph that the number of parcels that is tracked from a certain area and time step depends on the evaporation…or precipitation… from the respective location or area at the respective time step. Is the number of parcels 100 or does it vary? Would be good to make this paragraph a bit clearer to fully understand what's happening in the lagrangian model. Could consider adding in a schematic figure?

Although we agree that, in principle, surface release is more realistic, given the specific set-up of UTrack surface or profile release does not really matter. UTrack contains a random mixing parameter such that even with surface release, a parcel would be assigned a random vertical position within 24 hours anyway. Still, it may have some effect, which was tested by Tuinenburg & Staal (2020). From here, we can show that the difference in moisture transport distance between the two modes of release is considerably (about 4-5 times) less than the size of an individual grid cell in NorESM2.

The number of parcels per mm of evaporation (or precipitation in case of moisture tracking backward in time, but this is not applicable in the current study) can vary, because it is a parameter in the model framework. As default (Tuinenburg & Staal 2020, https://doi.org/10.5194/hess-24-2419-2020) it is 100 parcels per mm. Here, we chose to

release 1000 parcels per mm, but distributed across the globe. We will revise this paragraph for clarity.

You add further information under the 'Simulation settings' section. Perhaps these sections could be combined / adjacent for clarity?

We will consider this when revising and possibly restructuring our Methods.

Line119 – did you calculate vertically integrated moisture fluxes?

We did not, because of the added randomized vertical mixing in UTrack, which is done to compensate for known underestimation of vertical fluxes. This means that the vertical moisture fluxes in the forcing data lose most of their relevance. Indeed, including vertical moisture flows additional to the mixing scheme is optional in UTrack. However, the forcing data at multiple pressure layers are still very important to account for differences in horizontal wind speed and direction.

Again we will clarify this shortly and refer to the references in which the model framework was introduced.

Line 158 – can you add more information on this basin-level analysis. Do you calculate trajectories for multiple locations within each basin? How does this differ from the rest of your analysis. E.g. couldn't you just subset your existing trajectories you calculated for the whole globe? Add a statement on what additional insight this basin-level analysis provides.

In our revision, we will clarify the methods and merits of the basin-level runs. Subsetting existing trajectories is possible but not cost-effective, because it would have meant we needed many more parcels per run in the global analysis that additionally would have required much more information to keep track of.

Line 159 – 100 parcels vs 1000 parcels mentioned in line 152. Is that correct? If so, why the difference?

We will make it more clear in the revision that this decision is based on different areas of global versus basin masks.

Line 171 – add definition of evaporation recycling and how this differs from P recycling

We will do this.

Line 213 – here you discuss changes in precipitation, but the figure you point to (Fig. 3a) shows changes in precipitation recycling, which isn't the same thing. Maybe double check.

Thank you for pointing this out. Indeed, in this sentence, Figure A1 should be referred to, and Figure 3a only in the sentence following it.

Having read further on in the paper – you show the annual P changes in Figure A1. Would be good to direct to this in relevant parts of the text. In fact, it might be better to move A1 to

the main paper given its importance to the rest of your results – it will allow the reader to identify drying and wetting areas more easily. Make sure all supplementary figures are referenced somewhere so they don't get overlooked.

We understand this reasoning, but if allowed, we prefer to show in the main text only original results. We consider non-original results that facilitate understanding of the novel findings to be supplementary.

"In 45.4% of the grid cells that are projected to become drier (representing 1.1% of all land grid cells)" are these figures correct? Does this mean that in 1.1 % of land grid cells the drying is caused by a reduction in P recycling? This value seems quite low looking at the shaded areas in Fig 3, but perhaps that's because you're showing significant and no-significant changes, but only talking about the significant changes. Maybe consider rewording this paragraph and subsequent paragraphs or at least clarify in the text that only a very small land fraction shows statistically significant changes.

Exactly, this low number is because of the small area of significant drying in the SSP1-2.6 scenario: 0.087 * 0.272 * 0.454 = 0.011. In Figure 3, we show also non-significant areas, as the reviewer remarks. In the revision, we will make sure this will be clear to the reader.

Line 214 "…this drying is dominated by a decrease in the precipitation that originates from land." Point to a figure / evidence for this.

We will refer to Figure 3a here (as mentioned above in response to the point raised about referencing Figure A1).

Lines 211-245 I wonder if the information presented in these paragraphs would be better summarised in a table? The text gets a bit repetitive here and it might be more useful to focus the reader's attention on your most important findings, e.g. perhaps the worst case scenario of SSP5-8.5 could be highlighted in the text? Just an idea.

This is a useful suggestion which we will implement.

Line 248 – Are you considering 'all global grid cells' or just land? As all your figures show land only, perhaps it makes sense to only refer to results over land throughout your description of your results. Check here and elsewhere.

We meant all global land grid cells. Thank you for pointing this out. We will check throughout the text whether we did not confuse this elsewhere.

Line 255 – please indicate a figure after the first sentence so the reader immediately knows where to look for evidence for this statement.

We will refer to Figure 6 after the first sentence.

Line 264 – again please be clearer with your figure indications. How is it possible for recycling to exceed actual precipitation? You indicate the regions where this occurs but

would be useful to add a line to explain if this is simply an artefact of the model or if this is a physical result (and if so how that might arise).

We will add explanation of how this is possible. It is an artefact of the model, where sometimes (especially in dry grid cells and months) too many forward-tracked moisture parcels end up in a grid cell relative to the precipitation in that grid cell in that month. This can be due to the stochastic nature of the model, the fact that some parcels are tracked across two months, and/or due to a non-closed water balance in the forcing data.

We also will add in the Discussion that this robustness and associated artefacts could also be tested in the future with different members of the NorESM runs (i.e. replicate runs with similar forcings and scenarios but with small different initial conditions)

Line 270 – consider removing the word 'one' or even the whole phrase in brackets – not sure if it's needed here and sounds a bit confusing

We will remove 'one' as it is redundant.

Line 272 – 'In SSP2-4.5, global forest cover remains 25%' insert 'at' after 'remains'

Thank you, we will do that.

Line 279 – 'a larger proportion of the 26 major…'

Good suggestion.

Line 280 – unclear what the distinction is between basin precipitation recycling ratio and terrestrial precipitation recycling ratio. In line 165 you state: "We calculate the global terrestrial precipitation recycling ratio as the percentage of precipitation on land that evaporated from land." Can you also clarify in caption for Table 2?

We will be more clear about the distinction between basin precipitation recycling ratio and terrestrial precipitation recycling ratio in both the main text and the caption of Table 2.

Line 289 – "Both changes in basin recycling ratio increases in SSP1-2.6 are an increase." Not sure what you mean here. Possibly: "The two basins that showed statistically significant changes in basin recycling ratio by the end of the century in SSP1-2.6 both showed increases from the baseline"?

This was a typo, thank you for spotting it. The suggested revision captures the intended meaning.

Line 289 – From Table 2 it looks like the change is from 19 % to 20% for Amur? Similarly it looks like Ob changed from 11 % to 13 % ( not 11 to 12 as written in the text). Possibly double check these and other values referenced in the text in case the numbers have updated since an earlier draft of the paper.

Numbers have indeed changed from an earlier version of the manuscript. We appreciate the detailed checks by the reviewer and we will double-check all numbers before resubmitting the manuscript.

Table 2 – It might be nice to somehow indicate (possibly through additional columns, or by colouring the numbers red and blue) which basins were showing statistically significant increases and which decreases in P recycling. At present the reader can quickly pick out the values with asterisks next to them but not the direction of changes.

This is a good suggestion. We will explore options to show the differences between increases and decreases in a clear way.

Line 302 – check the wording here as it seems a bit contradictory. You say that Chad has an "increase in basin recycling ratio between the middle and end of the century" but in the next sentence you say "the Chad basin does not have a significant decrease by the end of the century". This whole paragraph might be a bit clearer if you first focus on the changes by the middle of the century, and then subsequently describe the changes that occur from mid to late century. I appreciate there is a lot of detail to try to capture, but at the moment it gets a bit muddled and the message gets lost.

Thank you for pointing this out. We will restructure the paragraph to improve its clarity.

Line 319 – I'm not familiar with the expression "the one percentage point level" which comes up here and elsewhere. Is the point you want to make that in some instances there are changes in P recycling where there are no changes in forest cover or cropland cover? Possibly rewording could improve clarity.

We will rephrase this. What we meant is that the ratios are rounded to percentages and there is no difference between those rounded numbers.

Line 321 – where land cover changes are small and P recycling changes are high, is this related to changes in e.g. plant stomatal conductance in response to rising atmospheric $CO_2$? Or other aspects of climate? I know this is only the results section, not results and discussion, but might be nice to just add in a line or so to briefly explain this finding.

It may be related to increased residence times of moisture in a warmer atmosphere, as explained in the Discussions section (lines 396-408). We will add a note about that also here in the Results.

Lines 317-330 Maybe restructure this paragraph or split into multiple paragraphs. For example, the first line of the paragraph really relates to what is described in the second half of the paragraph – i.e. regions where changes in recycling are related to changes in land cover

We will restructure and/or split it up.

Line 345 – missing word 'relative' for panel a description. Could start the paragraph with the current second sentence (rephrased!) and focus on areas with no LCC that see changes in

recycling, then have a separate paragraph that looks at areas that do see large differences in LCC.

Thank you, we will add "relative" to the caption.

We assume that the second part of the comment belongs to the previous comment about lines 317-330. We will take this into account when restructuring it.

Lines 332-336 Maybe reword. I think the point you are trying to get across is: "In the Amazon and Congo river basins, end-of-century recycling ratios are the same in SSPs with different land cover distributions. For example, in the Amazon under SSP 3-7.0 and SSP5-8.5 the estimated basin recycling ratio is 24% and the terrestrial recycling ratio is 40%, despite forest cover of 75% in SSP3-7.0 and 82% in SSP5-8.5. Similarly in the Congo…. etc. etc."

Thank you for this specific suggestion; it is indeed the point that we were trying to get across.

Line 349 – specify 'absolute' differences in ΔTPR

This figure displays the differences in relative recycling, not absolute, which we will specify in line 349.

Line 355 – missing word 'ratio' after recycling. Not sure if this is intended given the change in units from Figure 2 to Figure 3?

Figure 3 shows absolute recycling, not ratios. We will add "absolute" to prevent misunderstanding.

Line 420 – can you speculate on why the Danube might show opposite behaviour?

We will delete the sentence starting with "Interestingly, though", as the Danube is in fact not so different from the general pattern; it just has a particularly strong decrease in terrestrial precipitation recycling ratio.

Line 423 – consider citing papers by Christopher Skinner and Rob Chadwick here. E.g. DOI: https://doi.org/10.1175/JCLI-D-16-0603.1

Thank you, this is a useful suggested reference and we will look for possible other ones.

Line 444 – underestimated…. In the current generation of climate models? Or specifically in this study?

We mean in our study, which we will make more clear.

Line 461 "We call drying land-dominated if it coincides with a significant increase in terrestrial precipitation recycling ratio and we call it ocean-dominated if it coincides with a significant decrease in terrestrial precipitation recycling ratio." This definition could come earlier as it would help understand the description of these results.

We agree, this definition should be provided in the first paragraph of section 4.2.

Line 524 – maybe repeat here that NorESM2 was the only model that provided the required variables at the time frequency required for your moisture tracking model.

Thank you, we will do this.

---

## Author Comment (AC2)

This paper addresses the question how moisture recycling ratios change under different future SSP scenarios. The authors study this question by running the UTrack moisture tracking model forced with the NorESM climate model and analyse 10-years of future climate slices under different SSP scenarios. This study on moisture recycling changes towards the future is very relevant and timely, however I have some major comments on the methodology and the reporting of the results. In short, my major concerns are 1) regarding the fact that only 10-year simulations are analysed (10-years is not a climatology), 2) that the changes in the model are not well described and not validated, 3) that present climate simulations are not validated well with literature or ERA5, 4) that relevant literature is missing in the introduction, and 5) that the results section reads as a bookkeeping exercise (of which the information can be put into a table) rather than a story highlighting the main results. I have described my major concerns in more detail below, and have also included substantial and minor comments that I encountered while reading the paper.

We are happy to read that the reviewer finds our manuscript timely and relevant, and we thank them for the constructive and thorough comments. Below we respond to these comments in more detail and explain how we aim to deal with them in a revision.

Major comments

**10-years of simulations do not present a climatology**

The results of this study are based on moisture recycling ratios calculated for slices of 10years of climate data (present and future). In climate terms, this is a very short period to draw conclusions from, taking into account the internal variability of our climate. By only analysing 10 years, it could happen that your results are biased to multiple dry or wet years present in the data. While the authors address the fact that they do not study interannual variability (although presenting standard deviations around moisture recycling ratios, which is an indication of interannual variability), they do not acknowledge the short 10-year timeseries as serious constrain to base their conclusions on. Further they do not compare their 10-year land precipitation and land evaporation results with a wider range of models to validate if the 10-years is a good representation of (present) climatology.

It is true that a ten-year period may be too short for a climatology, because of variability in the climate. Still, we wanted to compare equally long periods with each other and we considered the first ten years of the simulations (2015-2024) representative of "the present". In the revision, we will explore the effects of using this ten-year benchmark. We expect either to increase the window or to present in the supplement additional results for the other ten-year windows if these results demonstrate that our ten-year period is long enough to account for interannual variability.

**No validation is performed for present climate and with the new model set-up**

To continue on the previous comment, one way to verify if present climate from the NorESM simulations is actually representing our current present climate, one could validate the moisture recycling ratios from NorESM baseline with moisture recycling ratios from ERA5.

The authors have published simulations with the UTrack model forced with ERA5, which is the perfect reference dataset for this study, and I am surprised to see no validation is done at all. Validation is recommended in two ways: a) to validate how well NorESM performs in representing moisture recycling ratios in current climate (2015-2024) and b) to validate the described model changes. With the latter I mean that a different model set-up is described based on the constrains of available data from NorESM and the impacts of using only limited input data (daily timestep, only eight pressure levels in the atmosphere). The impact of using daily data and limited information in the vertical (and horizontal) can be perfectly validated with the ERA5 dataset. One can run the UTrack model in the standard ERA5 set-up, and run the UTrack model with the input from ERA5 based on the constrains of NorESM, This will allow to illustrate the impacts of using limited data resources, which is currently not addressed in the paper at all. The fact that there are some grid cells that show higher precipitation recycling then precipitation itself (lines 264-269), might be related to the fact that only daily data or few vertical levels are used.

Thank you, we appreciate these suggestions. We agree that a "validation" – even though it would not be a true validation against observations – using ERA5-based global moisture recycling is valuable. Still, we want to stress that we are mainly interested in relative changes in precipitation recycling, exactly because of possible model biases. Having noted that, we can "validate" the global patterns of precipitation recycling patterns using ERA5-based results of Tuinenburg et al. 2020 (https://doi.org/10.5194/essd-12-3177-2020), which are also run for a ten-year period (2007-2018). In our revision, we will provide and discuss comparisons of the global precipitation recycling patterns from Tuinenburg et al. (2020) and the current study.

Regarding the different set-up of the ERA5-based model: we appreciate this suggestion and agree it would be very interesting, but it lies outside the scope of this paper. This model transformation is not easy to do, the runs would still be very data-heavy and computationally expensive, and it would deserve a study on its own as we would have to do a systematic and global sensitivity analysis of all model changes. In Tuinenburg & Staal (2020, https://doi.org/10.5194/hess-24-2419-2020), we did perform some sensitivity analyses on the ERA5-based model version that we may use to interpret possible differences between the ERA5- and NorESM-based runs.

This point is also interesting in relation to the sensitivity of the results to different ensemble members of NorESM (also see our reply to Ref. 1). We believe that extra sensitivity analyses and model comparisons are very important. Indeed, currently a moisture tracking model intercomparison study is being carried out by an international community of moisture recycling modelers. This effort which will shed light on the importance of these model assumptions for moisture recycling results.

In addition, daily data from NorESM is used to force the UTrack model (Line 118: has a temporal resolution of one day). I assume that for wind fields and specific humidity instantaneous data is used and I wonder at which timestep of the day this data is taken? This is not stated in the methods and influences your results. If instantaneous wind fields are taken at midnight, features like a low level jet will enhance moisture transport, compared to instantaneous wind fields taken at noon. Opposite, sea-breeze features which

enhance ocean-to-land moisture transport are mostly present during the day and thus will also influence your results when only one timestep during the day is used. This issue can be addressed by running UTrack with hourly ERA5 forcing, and with daily ERA5 forcing (as suggested in the previous point).

We agree that the daily resolution of the data from NorESM is a limitation. Diurnal fluctuations in winds, evapotranspiration and precipitation are thus averaged out, but it does not mean that only moisture flows at e.g. midnight or noon are accounted for. We will discuss this limitation more deeply in the revision.

Last, I am a bit confused why the authors use a forced climate scenario to analyse past climate. I can follow the logic to take the scenario that is following the trajectory that the world is currently on (Fricko et al., 2017) (line 165), though citing a paper from 7 years ago feels a bit odd then. Wouldn't it be much more logic to use climate data forced with observed CO2 levels and observed SSTs?

We decided to do this for internal consistency, allowing us to isolate the relative differences in precipitation recycling during this century. We agree that citing a paper from less than 7 years ago would be more appropriate and we will look for the most up-to-date reference for this.

**Introduction does not include all relevant literature and hypothesis**

In the introduction the authors state that (line 50): "However, where, how, and to which extent terrestrial moisture recycling will change in the future remains unclear." Although I agree there is still research to be done on how moisture recycling is changing to the future, there is also literature available already that addresses and (partly) answers this statement, and this literature is only cited in the discussion. An introduction needs to state the relevant literature on the topic and this is currently not done. Examples of literature that addresses the changes in moisture sources or moisture recycling in a future climate (Benedict et al., 2020; Findell et al., 2019; Fernandez-Alvarez et al., 2023). Furthermore, the introduction is also the moment to state hypothesis based on current literature, for example addressing the impacts of land-use change versus climate change. Currently the introduction provides more insights on the methods, describing the SSP scenarios and different moisture tracking methods which I found more relevant for the method section, or can be reduced.

We agree. We will make an effort to include the most relevant literature in the Introduction and we appreciate you pointing us to these important papers.

**Improve results and discussion section**

In the first section of the results (line 187-192) the absolute and relative changes in land precipitation and land evaporation are given from NorESM. It would be very good to put these numbers in the perspective of a multi-model mean, for example given in Table 8.1 of Chapter 8 of the IPCC 2021 report, Douville et al., (2021). This indicates if the precipitation and evaporation averages from NorESM2 fall within or outside of the range of the CMIP6 multi-model mean.

This is a good suggestion. We will provide this context in the revision.

The result section reads as a bookkeeping exercise, where multiple alinea's (four alinea's from line 210 to line 245) have exactly the same structure but different numbers inserted for the different scenario's. This makes the result section dry and hard to read. This information suits well for a table instead, while in the text rather the interesting findings of the table can be reported.

Also reviewer 1 suggested to include a table to summarize the results presented in lines 211-245. We will do this.

Further, my suggestion would be to combine the results and discussion section to allow for direct comparison with literature. Currently, the result section is very dry as it is a sum-up of numbers and scenario's. By directly comparing moisture recycling ratios with the literature (combining results and discussion) allows for more perspective. At the moment, in the discussion the numbers from the result section are not repeated, which makes it very hard to put literature results next to the results of this study, which I think is very important to do. The same holds for the results on the major river basins, which could include more references to current literature.

We need to comply to journal requirements regarding paper structure, but we take this suggestion at heart and will reorganize and rewrite the Results section to improve its readability. Also reviewer 1 provided a lot of useful suggestions in this regard.

On the discussion on the impact of land-use change and climate change on recycling ratio. I think this is a very interesting discussion point which is now addressed only shortly, I would dedicate a whole section on this. Are their different ways forward to test this influence? Some arguments that are provided later in the text (line 411) could potentially also be used to study impact land-use change vs climate change.

This is a good suggestion. We will elaborate on this and propose ways forward to test the influence of climate change versus land-cover changes, likely in a separate section.

**Substantial comments**

Line 40-50: Besides including relevant literature that assessed moisture recycling under a warming climate it is also good to address the impact of circulation changes on moisture recycling, such as changes in location of the ITCZ, Hadley cells, storm tracks, as this will affect the moisture transport as well.

We will add reference to circulation changes in the Introduction.

Section 2.3 Simulations settings; I already addressed the issue of daily data in the major comments, but the limitation of only having limited pressure levels as input is not discussed at all. What is the impact of this on your results?

We believe the number of pressure layers (eight) is quite large, although it is smaller than in ERA5. See also the earlier comment. A systematic assessment of this effect is beyond the

scope of this study, but Tuinenburg & Staal 2020 (https://doi.org/10.5194/hess-24-2419-2020) did do some tests and found that severely degrading the vertical moisture profile can have substantial effects, affecting moisture transport distances in the order of hundreds of km. We will include a discussion of this in our revision.

Line 176-184: Can the significant increases and decreases be quantified? Did you use a threshold to call it a significant increase or decrease?

Yes, we used $\alpha=0.05$. Thanks for pointing out this missing information.

Section 2.4: When is the model initialized? As already mentioned I am a bit surprised that for current climate a scenario is used, while we already have the observations of current climate as the boundary conditions of the model. I could imagine this is done for consistency, but it would be nice to check how well the baseline run represent the actual conditions. Further, are these atmosphere only runs? So SST is prescribed?

Indeed, we do this for consistency. As explained above, we will use the results from Tuinenburg et al. (2020) to compare the global patterns of precipitation recycling against. The NorESM runs were coupled, without prescriptions of SST (Seland et al. 2020, https://doi.org/10.5194/gmd-13-6165-2020).

Line 174: Do I understand correctly that also for the SSP3-7 and SSP1-2.6 and SSP 3-7.0 you use the same climate sensitivity as SSP5-8.5? The approach here is unclear but you would expect that the climate sensitivity is used per SSP scenario to calculate changes in moisture recycling per degree warming.

We based it on only SSP5-8.5, because it has the largest increase in temperature, allowing for greater accuracy. We will clarify this.

Further, we are currently already warmer than the 1976-2005 baseline that is mentioned for which the 3.26 degrees is determined for. In the results a 3.26 change in temperature is used to move from the SSP2-4.5 (baseline; 2015-2024) to the SSP5-8.5, but I assume there is already some warming in the SSP2-4.5 baseline run for 2015-2024, which is now not taken into account. Thus if I understand the taken approach well, using the 3.26 degrees warming is incorrect.

This is correct and it may lead to a slight error. We will improve this calculation by correcting for warming since the 1976-2005 baseline.

Results

Line 177: "wetting, land dominated" if a significant increase in precipitation coincides with a significant **decrease** in terrestrial precipitation recycling --> should this not be exactly opposite? An increase in terrestrial precipitation recycling? Line 461-462 also states that land-dominated means increase in terrestrial precipitation recycling

Thank you for spotting this typo, it should indeed be the opposite.

I would suggest to leave the min and max value of moisture recycling out of the text to make it more readable. Those min and max values could be reported in a table. By providing the std you give an idea of the interannual variability in the text.

We agree that the minima and maxima can also be included in Table 1.

Line 213-215; How relevant is it to give this information if it only concerns such a small percentage of land grid cells (1.1% and 1.3%)? Instead, it would be nice if some words are dedicated on where those 8.7% of land grid cells are on the globe that show a significant change in precipitation. And do you mean with precipitation absolute precipitation or precipitation recycling? In line 170 it is stated that statistical significance is tested for precipitation recycling, but from the result section it reads as if it is checked for absolute precipitation changes, which is confusing. I read in the caption of Figure 2 and 3 it is about significant differences in precipitation recycling, it would be nice to have those regions that show a significant change are hatched in figure 2 and 3.

We believe that even if it represents a quite small area of the global land, these numbers are still relevant to report. We agree, though, that some more detail about where the areas with significant changes are located will be a worthwhile addition. We will also try to clarify to solve mentioned confusion.

Line 279-277: I am not sure about the purpose of this alinea. Are these findings of this study? Or are these numbers given here to indicate the impact of land-use change on moisture recycling? If so, I would discuss them in combination with the discussion section 4.1.

These are numbers we calculated, but in essence are not novel results. Indeed we provide these numbers so we can better interpret the land-cover versus climate change effects on moisture recycling. In our restructuring of the Results section we will take this point into account.

**Minor comments / typos / small unclarities:**

Define precipitation recycling in abstract

OK.

Line 20: moisture recycling ratio--> do you mean with moisture recycling precipitation recycling? Terms are used throughout it each other but it is unclear what is what

Thank you, we will be more clear.

Line 108: For equations  --> equations of what? Of the moisture tracking model?

Yes. We will specify this.

Line 134: 'There is some overestimation of global mean temperature' --> this is very vaguely stated, can you quantify?

We will be more specific based on the referenced paper.

Line 148: 'We used these forcing data directly without interpolation' --> How can you have daily data and run the model on 4-hourly timesteps, without interpolation?

Individual moisture parcels may cross multiple grid cells during one time step if the time step is too large. This may cause errors in the parcel trajectories, which is solved by taking a sufficiently small time step, even if the data themselves are not interpolated.

Line 159: In Line 153 it says 1000 parcels per mm, and here 100 parcels per mm

Correct. We used different settings for the global runs and the basin runs, based on the fact that the areas of the two differ by orders of magnitude. Effectively, the 100 parcels per mm in the basin runs mean that the same parcel already represents a larger volume of water in the basin runs compared to the 1000 parcels in the global run.

3.1 Global land --> can you make the headers more self-explanatory? The result section will also benefit from more section and section headings to illustrate the red-threat

We agree it would be good to make the headings more self-explanatory.

Line 492: less clouds but more rain? Maybe I misunderstand

We will delete "likely linked to underestimated cloud cover", as the point simply is that precipitation over oceans is overestimated in NorESM2.

Line 499: 'Around one-fifth of global precipitation is attributed to vegetation' --> to me it is not clear what is meant with this sentence

We will rephrase. What is meant is that vegetation, through enhanced evapotranspiration, is estimated to be responsible for one-fifth of global precipitation, that is, without vegetation global precipitation would decrease by that amount.

Conclusions

Line 538-539: 'widespread drying accompanied by disproportional reductions of moisture supply over land' à this sentence counteracts the argument of global greening and increased evaporation (stated in the discussion)

We will be more clear that here regional-scale drying is meant, such as in the Amazon and eastern Europe, which contrasts with the global average.

Here the word disproportional is often used, what is meant with that?

We mean that the relative change in moisture supply exceeds the relative change in precipitation. We will make sure this is clear to the reader.

Line 541-542: can you back-up this last sentence by findings from the study?

This is a high-level take-away based on the discussion of our results in the context of the literature. We will make this more explicit.

Figures

Figure 2 to 6: These figures can be improved and made more readable by only displaying one legend (colorbar) per figure, and not for all subfigures. In this way the figures can be enlarged.

Good suggestion, we will do that.

---

## Author Response (AR1)

Dear authors,

Both reviewers find merits with your study, yet also highlight shortcomings. Based on their comments, and your replies, I am returning the manuscript for major revisions. With the support of the Reviewers, I will then re-evaluate the revised manuscript's suitability for publication in ESD.

From my own reading, of your manuscript and the reviews, I suggest substantial changes in the following:

- Model validation UTrack. References to published papers can of course be used, but it seems you have made some important changes to model parameters (time step, vertical levels). Please validate how these changes impact model performance.

- Forced signal vs interannual variability. Though somewhat dependent on the variable of interest, comparisons of 10 year periods are subject to substantial interannual variability. Please analyse the robustness of your results. Note some information might be present in your collection of future runs (4 SSP, 2 time horizons), if path dependence is small these should broadly scale with global warming levels.

- Given that the results are based on a single GCM, I suggest to add "using NorESM" in the title

Best regards,
Karin van der Wiel

Dear Dr. Van der Wiel,

Thank you for the opportunity to revise our manuscript. We also thank both reviewers for their thorough feedback, which helped improve our manuscript. We reply to the individual comments below, but regarding the three points mentioned above:

- We now validate the model output (from the baseline period) using previous global runs of the original ERA5-based UTrack model. We use the same major river basins as used in Table 2 and compare their terrestrial precipitation recycling ratios from the two models. We find that the global patterns are mostly consistent, but observe a systematic underestimation of the NorESM2-based runs compared to ERA5: by comparing the basin recycling ratios of the 26 major river basins of the world, we see that the estimates based on NorESM2 are on average 9.4 percentage point lower than those from Tuinenburg et al. (2020). The absolute differences are on average 9.8 percentage point (see also Fig. A6). The fact that the average absolute differences are very similar to the average differences shows that the bias is systematic; only for two out of 26 river basins, the NorESM2-based estimates are slightly larger. Because we are primarily interested in relative changes in recycling (and the differences among SSPs therein), we believe a systematic bias like this is acceptable for our purposes. Furthermore, we found that NorESM2 is in the middle

of the pack among CMIP6 models regarding projected hydrological changes with global warming.

- We explored the interannual variability in global precipitation recycling ratio using both 10-year and 30-year windows. We find that using 10-year windows does lead to larger fluctuations in estimated moisture recycling, but that the bandwidth of this variability (mean ± one standard deviation) is very similar to that using 30-year windows.

- We think that adding the specific "NorESM" (or "NorESM2") in the title would also warrant including "UTrack" in the title, as that is the actual model that we used for moisture tracking. We prefer not to make the title too method-oriented, but emphasize the differences among SSPs. Therefore, if it is allowed, we would prefer to maintain the title as it is.

Please note that when we refer to line numbers in our responses, they apply to the manuscript version with tracked changes.

We hope that our revised manuscript is now considered suitable for publication in Earth System Dynamics.

On behalf of all authors,

Kind regards,

Arie Staal

**R1**

**Summary**

The authors use future projections from the NorESM2 climate model to drive the atmospheric moisture tracking model UTrack. The future projections represent four possible socioeconomic scenarios of varying severity, with the aim of estimating global responses in terrestrial precipitation recycling to different emissions pathways. The authors present their new version of the UTrack model in a Github repository. The paper is well written and a valuable addition to the literature. The introduction and discussion sections were especially good, however, the description of the results could be improved to help the reader make links between the text and the figures more readily, and to ensure your findings come across in the clearest possible way.

We thank the reviewer for their kind words and the constructive and detailed comments. They helped us improve the manuscript, in particular the clarity and readability of the Results. Below we respond briefly to the individual comments and how we dealt with them.

Line by line comments

Line 74 – add some examples of studies that use moisture tracking models to assess precipitation recycling

We added several examples, from local to global examples, and with different purposes (lines 79-82 in version with tracked changes): "These models are used to study moisture

recycling on different scales and with different purposes: for example, the upwind water supplies of cities (Keys et al., 2018), the effects of land cover changes on global breadbaskets (Bagley et al., 2012), the role of tropical forests in maintaining their own rainfall levels (Staal et al., 2020b), and to assess global patterns of continental recycling (Van der Ent et al., 2010)."

Line 97 – can you add further explanation for why 100 parcels are released at random locations above the starting area? For evaporation, shouldn't parcels come from the surface, and precipitation higher up in the atmosphere? You say further up in the paragraph that the number of parcels that is tracked from a certain area and time step depends on the evaporation…or precipitation… from the respective location or area at the respective time step. Is the number of parcels 100 or does it vary? Would be good to make this paragraph a bit clearer to fully understand what's happening in the lagrangian model. Could consider adding in a schematic figure?

Although we agree that, in principle, surface release is more realistic, given the specific set-up of UTrack surface or profile release does not really matter. Most important reason in here it that UTrack contains a random mixing parameter such that even with surface release, on average a parcel would be assigned a random vertical position within 24 hours anyway. We have tested this effect in Tuinenburg & Staal (2020) and from here we can conclude that the effect on transport distance is about 4-5 times less than the size of an individual grid cell in NorESM2.

The number of parcels per mm of evaporation (or precipitation in case of moisture tracking backward in time, but this is not applicable in the current study) can vary, because it is a parameter in the model framework. As default (Tuinenburg & Staal 2020, https://doi.org/10.5194/hess-24-2419-2020) it is 100 parcels per mm. Here, we chose to release 1000 parcels per mm, but distributed across the globe. We removed from this paragraph the reference to 100 parcels per mm and revised the text as follows (lines 104-105): "At every time step, a number of parcels per mm evaporation or precipitation (to be defined by the user) is released at random locations above the starting area."

You add further information under the 'Simulation settings' section. Perhaps these sections could be combined / adjacent for clarity?

We considered this, but believe it is more logical to first introduce the data that we used in this study before explaining the simulation settings, which depend on the details of the forcing data.

Line119 – did you calculate vertically integrated moisture fluxes?

The UTrack model does not make use of the vertically integrated moisture flux directly, which is different from alternative moisture recycling models (such as WAM2-Layers; Van der Ent et al. 2014) that do use this as input variable. In UTrack, the horizontal moisture transport is implicitly calculated by the Lagrangian scheme. In a nutshell, we simulate the atmospheric moisture transport by quantising the atmospheric water column as a large number of moisture parcels at different altitudes. These parcels are transported with the winds at the relevant altitudes to arrive at a new location at each time step. This will give an updated location for each parcel at every tilmestep, from which a vertical moisture profile could be constructed. This difference between the vertical moisture profiles between the

time steps could be used to determine the implied divergence/convergence and thus the vertically integrated moisture fluxes, but we do not actually do this within the scheme.

We also added the following related revision: "The wind speeds and vertically integrated tendency of air pressure are calculated for eight pressure levels: 1000 hPa, 850 hPa, 700 hPa, 500 hPa, 250 hPa, 100 hPa, 50 hPa, and 10 hPa. Horizontal fluxes are purely based on these pressure levels, whereas vertical fluxes additionally include the above-mentioned probabilistic parcel repositioning every 24 hours (Tuinenburg & Staal, 2020)." (lines 128-131)

Line 158 – can you add more information on this basin-level analysis. Do you calculate trajectories for multiple locations within each basin? How does this differ from the rest of your analysis. E.g. couldn't you just subset your existing trajectories you calculated for the whole globe? Add a statement on what additional insight this basin-level analysis provides.

We added the following information (lines 174-182): "In addition to global recycling ratios, we were interested in basin precipitation recycling ratios for individual river basins located across the globe. Analogous to global terrestrial precipitation recycling, we calculated basin precipitation recycling ratios as the percentage of precipitation that originated as evaporation from the same basin. Because the global runs involved simultaneous parcel tracking from everywhere across the globe, which do not allow for calculating basin recycling, we required separate runs for this. Therefore, we performed forward-tracking runs for the 26 major river basins of the world using shapefiles from the Global Runoff Data Centre (GRDC, 2020). We performed these runs again for all SSPs. Instead of 1000 parcels for every mm of evaporation globally, we released 100 parcels for every mm of evaporation from each basin. Note that this implies a larger amount of parcels per volume than in the global runs due to the considerably smaller source area of the basins."

Line 159 – 100 parcels vs 1000 parcels mentioned in line 152. Is that correct? If so, why the difference?

We clarified this point in the revision (see previous reply).

Line 171 – add definition of evaporation recycling and how this differs from P recycling

We revised to: "We also report global evaporation recycling, the percentage of terrestrial evaporation that precipitates over land. Calculating evaporation recycling is possible given that the source area of the tracking equals the target area (the global land area)." (lines 192-194)

Line 213 – here you discuss changes in precipitation, but the figure you point to (Fig. 3a) shows changes in precipitation recycling, which isn't the same thing. Maybe double check.

Thank you for pointing this out. Indeed, in this sentence, Figure A1 should be referred to, and Figure 3a only in the sentence following it (and similarly in the following paragraphs).

Having read further on in the paper – you show the annual P changes in Figure A1. Would be good to direct to this in relevant parts of the text. In fact, it might be better to move A1 to the main paper given its importance to the rest of your results – it will allow the reader to identify drying and wetting areas more easily. Make sure all supplementary figures are referenced somewhere so they don't get overlooked.

We understand this reasoning, but we prefer to show in the main text only original results. We consider non-original results that facilitate understanding of the novel findings to be supplementary. We added reference to Figure A1 in line 310.

"In 45.4% of the grid cells that are projected to become drier (representing 1.1% of all land grid cells)" are these figures correct? Does this mean that in 1.1 % of land grid cells the drying is caused by a reduction in P recycling? This value seems quite low looking at the shaded areas in Fig 3, but perhaps that's because you're showing significant and no-significant changes, but only talking about the significant changes. Maybe consider rewording this paragraph and subsequent paragraphs or at least clarify in the text that only a very small land fraction shows statistically significant changes.

Exactly, this low number is because of the small area of significant drying in the SSP1-2.6 scenario: 0.087 * 0.272 * 0.454 = 0.011. In Figure 3, we show also non-significant areas, as the reviewer remarks. We clarified by adding "(see Fig. 3, including non-significant changes)" in lines 312-313.

Line 214 "…this drying is dominated by a decrease in the precipitation that originates from land." Point to a figure / evidence for this.

We now refer to Figure 3 in line 312.

Lines 211-245 I wonder if the information presented in these paragraphs would be better summarised in a table? The text gets a bit repetitive here and it might be more useful to focus the reader's attention on your most important findings, e.g. perhaps the worst case scenario of SSP5-8.5 could be highlighted in the text? Just an idea.

Thank you for this suggestion. We summarized these results in a new supplementary Table (Table A4, below) and now summarize the main findings in the main text, highlighting the results for SSP5-8.5, which are the most extreme ones (lines 304-308):

"With a more severe SSP, the proportion of global land that experiences a significant change in precipitation by the 2090s increases, from 8.7% of global land cells in SSP1-2.6 to 41.5% in SSP5-8.5. Whether this change in precipitation is drying or wetting, we find a larger proportion of land grid cells in which terrestrial precipitation recycling ratio decreases as the SSP becomes more severe (Table A4)."

Table A4: The percentage of global terrestrial grid cells with a significant change in annual precipitation, its direction of change (drying or wetting), and its direction of change in terrestrial precipitation recycling (TPR) ratio for each SSP by the 2090s compared to the baseline period.

|  | SSP1-2.6 | SSP2-4.5 | SSP3-7.0 | SSP5-8.5 |
|---|---|---|---|---|
| Significant change in precipitation | 8.7% | 17.1% | 30.0% | 41.5% |
| of which drying | 27.2% | 22.7% | 16.4% | 19.0% |
| of which increasing TPR ratio | 56.6% | 55.7% | 30.0% | 24.5% |
| of which decreasing TPR ratio | 45.4% | 44.3% | 70.0% | 75.5% |
| of which wetting | 72.8% | 77.3% | 83.6% | 81.0% |
| of which increasing TPR ratio | 62.1% | 50.1% | 43.5% | 32.9% |

| of which decreasing TPR ratio | 37.9% | 49.9% | 56.5% | 67.1% |

Line 248 – Are you considering 'all global grid cells' or just land? As all your figures show land only, perhaps it makes sense to only refer to results over land throughout your description of your results. Check here and elsewhere.

We meant all global land grid cells. Thank you for pointing this out. We fixed it throughout the manuscript.

Line 255 – please indicate a figure after the first sentence so the reader immediately knows where to look for evidence for this statement.

We now refer to Figure 6 after the first sentence (line 327).

Line 264 – again please be clearer with your figure indications. How is it possible for recycling to exceed actual precipitation? You indicate the regions where this occurs but would be useful to add a line to explain if this is simply an artefact of the model or if this is a physical result (and if so how that might arise).

We revised the paragraph as follows: "In some grid cells, estimated precipitation recycling exceeds precipitation itself due to model artifacts. In the baseline scenario, this occurs in 1.3% of global land grid cells. These areas are depicted as having a precipitation recycling ratio of 100% in Fig. 1a and are mainly located in the Himalaya and the Andes mountains. Sometimes, too many forward-tracked moisture parcels end up in a grid cell relative to the precipitation in that grid cell in that month. This can be due to the stochastic nature of the model, the fact that parcels can be tracked across two months with a different water balance." (lines 326-343)

We also added in the Discussion that this robustness and associated artefacts could also be tested in the future with different members of the NorESM2 runs (lines 629-634):

"Furthermore, the model generates internal climate variability. Three ensemble members of historical runs gave around 0.1 °C difference in global temperature on a yearly basis (Seland et al., 2020). Such ensemble runs allow for the separation of internal climate variability from the forced climatic signal. In the future, different ensemble members can be used to force UTrack, so the signal and noise in precipitation recycling can be separated, even though it is unlikely that the global trends and patterns would be affected by different ensemble members."

Line 270 – consider removing the word 'one' or even the whole phrase in brackets – not sure if it's needed here and sounds a bit confusing

We removed the phrase in brackets.

Line 272 – 'In SSP2-4.5, global forest cover remains 25%' insert 'at' after 'remains'

Thank you, we fixed it.

Line 279 – 'a larger proportion of the 26 major…'

Good suggestion, we implemented it.

Line 280 – unclear what the distinction is between basin precipitation recycling ratio and terrestrial precipitation recycling ratio. In line 165 you state: "We calculate the global terrestrial precipitation recycling ratio as the percentage of precipitation on land that evaporated from land." Can you also clarify in caption for Table 2?

We added the following sentence in lines 175-176: "Analogous to global terrestrial precipitation recycling, we calculated basin precipitation recycling ratios as the percentage of precipitation that originated as evaporation from the same basin." We also clarified it in the caption of Table 2: "Basin recycling ratio is the percentage of precipitation in a basin that has evaporated from the same basin; terrestrial precipitation recycling ratio is the percentage of precipitation on land that has evaporated from land."

Line 289 – "Both changes in basin recycling ratio increases in SSP1-2.6 are an increase." Not sure what you mean here. Possibly: "The two basins that showed statistically significant changes in basin recycling ratio by the end of the century in SSP1-2.6 both showed increases from the baseline"?

This was a typo, thank you for spotting it. The suggested revision captures the intended meaning (lines 365-366).

Line 289 – From Table 2 it looks like the change is from 19 % to 20% for Amur? Similarly it looks like Ob changed from 11 % to 13 % ( not 11 to 12 as written in the text). Possibly double check these and other values referenced in the text in case the numbers have updated since an earlier draft of the paper.

Numbers have indeed changed from an earlier version of the manuscript. We appreciate the detailed checks by the reviewer and we double-checked all numbers and corrected where applicable.

Table 2 – It might be nice to somehow indicate (possibly through additional columns, or by colouring the numbers red and blue) which basins were showing statistically significant increases and which decreases in P recycling. At present the reader can quickly pick out the values with asterisks next to them but not the direction of changes.

This is a good suggestion. However, coloured table cells are not allowed according to journal requirements. Additional columns we did not prefer, in order to keep the Table as simple as possible.

Line 302 – check the wording here as it seems a bit contradictory. You say that Chad has an "increase in basin recycling ratio between the middle and end of the century" but in the next sentence you say "the Chad basin does not have a significant decrease by the end of the century". This whole paragraph might be a bit clearer if you first focus on the changes by the middle of the century, and then subsequently describe the changes that occur from mid to late century. I appreciate there is a lot of detail to try to capture, but at the moment it gets a bit muddled and the message gets lost.

Thank you for pointing this out. Indeed this paragraph was not easy to understand, because of significant versus insignificant increases and decreases for individual basins in different periods. We decided to remove the details about the significance for individual basins in particular periods and believe the paragraph is more clear now.

Line 319 – I'm not familiar with the expression "the one percentage point level" which comes up here and elsewhere. Is the point you want to make that in some instances there are changes in P recycling where there are no changes in forest cover or cropland cover? Possibly rewording could improve clarity.

What we meant is that the ratios are rounded to percentages and there is no difference between those rounded numbers. We removed the term throughout the manuscript.

Line 321 – where land cover changes are small and P recycling changes are high, is this related to changes in e.g. plant stomatal conductance in response to rising atmospheric $CO_2$? Or other aspects of climate? I know this is only the results section, not results and discussion, but might be nice to just add in a line or so to briefly explain this finding.

It may be related to increased residence times of moisture in a warmer atmosphere, as explained in the Discussions section (lines 491-502). We also added a sentence in lines 401-402: "This can be explained by increased residence times of moisture in a warmer atmosphere, increasing the typical distance that moisture travels before precipitating."

Lines 317-330 Maybe restructure this paragraph or split into multiple paragraphs. For example, the first line of the paragraph really relates to what is described in the second half of the paragraph – i.e. regions where changes in recycling are related to changes in land cover

We restructured this and the following paragraph (lines 395-425) and believe the story is told in a more logical and insightful way.

Line 345 – missing word 'relative' for panel a description. Could start the paragraph with the current second sentence (rephrased!) and focus on areas with no LCC that see changes in recycling, then have a separate paragraph that looks at areas that do see large differences in LCC.

Thank you, we added "relative" to the caption.

We assume that the second part of the comment belongs to the previous comment about lines former lines 317-330. We took this into account when restructuring the section.

Lines 332-336 Maybe reword. I think the point you are trying to get across is: "In the Amazon and Congo river basins, end-of-century recycling ratios are the same in SSPs with different land cover distributions. For example, in the Amazon under SSP 3-7.0 and SSP5-8.5 the estimated basin recycling ratio is 24% and the terrestrial recycling ratio is 40%, despite forest cover of 75% in SSP3-7.0 and 82% in SSP5-8.5. Similarly in the Congo…. etc. etc."

Thank you for this specific suggestion; it is indeed the point that we were trying to get across. We adopted this in the revision of the paragraph (lines 413-425).

Line 349 – specify 'absolute' differences in ΔTPR

This figure displays the differences in relative recycling, not absolute, which we specified in line 441 and similarly for absolute recycling in line 447.

Line 355 – missing word 'ratio' after recycling. Not sure if this is intended given the change in units from Figure 2 to Figure 3?

Figure 3 shows absolute recycling, not ratios. We added "absolute" to prevent misunderstanding.

Line 420 – can you speculate on why the Danube might show opposite behaviour?

We deleted the sentence starting with "Interestingly, though", as the Danube is in fact not so different from the general pattern; it just has a particularly strong decrease in terrestrial precipitation recycling ratio.

Line 423 – consider citing papers by Christopher Skinner and Rob Chadwick here. E.g. DOI: https://doi.org/10.1175/JCLI-D-16-0603.1

Thank you for this suggestion. We believe this to be a useful reference in lines 542-543: "The remaining 45% mostly resulted from the physiological effects of $CO_2$ increase rather than deforestation directly (Li et al., 2023b; see also Skinner et al., 2017)."

Line 444 – underestimated…. In the current generation of climate models? Or specifically in this study?

We mean in our study, which we added in line 551.

Line 461 "We call drying land-dominated if it coincides with a significant increase in terrestrial precipitation recycling ratio and we call it ocean-dominated if it coincides with a significant decrease in terrestrial precipitation recycling ratio." This definition could come earlier as it would help understand the description of these results.

We agree, so we moved this definition to the first paragraph of section 4.2.

Line 524 – maybe repeat here that NorESM2 was the only model that provided the required variables at the time frequency required for your moisture tracking model.

Thank you for this suggestion. We added in lines 636-637 the following: "…, since it was the only one providing the required variables at high temporal and spatial resolution for the Tier 1 scenarios".

**R2**

This paper addresses the question how moisture recycling ratios change under different future SSP scenarios. The authors study this question by running the UTrack moisture tracking model forced with the NorESM climate model and analyse 10-years of future climate slices under different SSP scenarios. This study on moisture recycling changes towards the future is very relevant and timely, however I have some major comments on the methodology and the reporting of the results. In short, my major concerns are 1) regarding the fact that only 10-year simulations are analysed (10-years is not a climatology), 2) that the changes in the model are not well described and not validated, 3) that present climate simulations are not validated well with literature or ERA5, 4) that relevant literature is missing in the introduction, and 5) that the results section reads as a bookkeeping exercise (of which the information can be put into a table) rather than a story highlighting the main results. I have described my major concerns in more detail below, and have also included substantial and minor comments that I encountered while reading the paper.

We are happy to read that the reviewer finds our manuscript timely and relevant, and we thank them for the constructive and thorough comments. Below we respond to these comments in more detail and explain how we dealt with them in the revision.

Major comments

**10-years of simulations do not present a climatology**

The results of this study are based on moisture recycling ratios calculated for slices of 10years of climate data (present and future). In climate terms, this is a very short period to draw conclusions from, taking into account the internal variability of our climate. By only analysing 10 years, it could happen that your results are biased to multiple dry or wet years present in the data. While the authors address the fact that they do not study interannual variability (although presenting standard deviations around moisture recycling ratios, which is an indication of interannual variability), they do not acknowledge the short 10-year timeseries as serious constrain to base their conclusions on. Further they do not compare their 10-year land precipitation and land evaporation results with a wider range of models to validate if the 10-years is a good representation of (present) climatology.

It is true that a ten-year period may be too short for a climatology, because of variability in the climate. Still, we wanted to compare equally long periods with each other and we considered the first ten years of the simulations (2015-2024) representative of "the present". We now explored the effects of using this ten-year time slice versus a thirty-year time slice. The figure below shows for each of the scenarios the mean ± one standard deviation of the global precipitation recycling ratio. Indeed we see some interdecadal variability of the mean around the trends, as indicated by the red lines. However, this variability remains well within the variability based on the thirty-year time slices across the scenarios. Therefore, we consider using the ten-year time slices justified, which allows us to study the trend along a longer time period and have a presumably more accurate representation of "the present". Also, the use of ten-year time slices is not unusual in CMIP6 (Mitchell et al. 2016, Nature Climate Change, https://doi.org/10.1038/nclimate3055). Furthermore, it allows for a better "validation" of the global precipitation recycling ratios (see our reply to the next comment).

We added the following text in lines 223-228:

"We also evaluated the choice of a ten-year time slice for our analysis. For this, we plotted the moving averages ± one standard deviation of the global terrestrial precipitation recycling ratios based on a ten-year time slice and a 30-year time slice, for each of the SSPs. We found that these ratios and their standard deviations largely overlap, indicating that ten-year time slices tend to be sufficient to capture most of the interannual variability in global precipitation recycling. Furthermore, especially in the severe scenarios SSP3-7.0 and SSP5-8.5 the trend in recycling ratio exceeds its variability, implying that long-term climate variability in NorESM2 does not affect our main outcomes (Fig. A7)."

[Figure]

Figure A7: Moving averages of the global terrestrial precipitation recycling ratio (TPR, in %) ± one standard deviation for ten-year time slices (in red) and 30-year time slices (in blue), for (a) SSP1-2.6, (b) SSP2-4.5, (c) SSP3-7.0, and (d) SSP5-8.5. Results for 2015–2099 were used, meaning that the moving averages for ten-year time slices range between 2020–2095 and those for 30-year time slices between 2030–2085.

**No validation is performed for present climate and with the new model set-up**

To continue on the previous comment, one way to verify if present climate from the NorESM simulations is actually representing our current present climate, one could validate the moisture recycling ratios from NorESM baseline with moisture recycling ratios from ERA5.

The authors have published simulations with the UTrack model forced with ERA5, which is the perfect reference dataset for this study, and I am surprised to see no validation is done at all. Validation is recommended in two ways: a) to validate how well NorESM performs in representing moisture recycling ratios in current climate (2015-2024) and b) to validate the described model changes. With the latter I mean that a different model set-up is described based on the constrains of available data from NorESM and the impacts of using only limited input data (daily timestep, only eight pressure levels in the atmosphere). The impact of using daily data and limited information in the vertical (and horizontal) can be perfectly validated with the ERA5 dataset. One can run the UTrack model in the standard ERA5 set-up, and run the UTrack model with the input from ERA5 based on the constrains of NorESM, This will allow to illustrate the impacts of using limited data resources, which is currently not

addressed in the paper at all. The fact that there are some grid cells that show higher precipitation recycling then precipitation itself (lines 264-269), might be related to the fact that only daily data or few vertical levels are used.

Thank you, we appreciate these suggestions. We agree that a "validation" – even though it would not be a true validation against observations – using ERA5-based global moisture recycling is valuable. Still, we want to stress that we are mainly interested in relative changes in precipitation recycling, exactly because of possible model biases. Having noted that, we can "validate" the global patterns of precipitation recycling patterns using ERA5-based results of Tuinenburg et al. 2020 (https://doi.org/10.5194/essd-12-3177-2020), which are also run for a ten-year time slice (2008-2017).

We now provide two new figures in the supplementary materials, which are also provided below. The global maps of precipitation recycling (Fig. A5) are generally consistent, but especially in the boreal zones the estimates from NorESM2 tend to be lower. This is also confirmed in Figure A6, which shows the precipitation recycling ratios of the 26 major river basins of the world according to both model versions. We decided to relate the recycling ratios for river basins rather than all global grid cells because of noise on a grid cell-by grid cell level, which should be smoothed out at the basin scale.

We added the following text in lines 210-221:

"To evaluate our model results against the literature, we compared precipitation recycling ratios from NorESM2 with UTrack simulations based on ERA5 for 2008–2017 (Tuinenburg and Staal, 2020). The global patterns are qualitatively similar, but the NorESM2-based estimates in high-latitude boreal zones are relatively low compared to those based on ERA5 (Fig. A5). We used the ERA5-based basin recycling ratios for the 26 major river basins as reported by Tuinenburg et al. (2020) for quantitative comparisons. We performed regressions between these basin recycling ratios and those from our baseline period. The reason we evaluated based on basin recycling rather than grid-cell-by-cell is the different spatial resolution between model versions and expected noise at relatively small spatial scales. The estimates based on NorESM2 are on average 9.4 percentage point lower than those from Tuinenburg et al. (2020). The absolute differences are on average 9.8 percentage point (Fig. A6). The fact that the average absolute differences are very similar to the average differences shows that the bias is systematic; only for two out of 26 river basins, the NorESM2-based estimates are slightly larger. Because we are primarily interested in relative changes in recycling (and the differences among SSPs therein), we believe a systematic bias like this is acceptable for our purposes."

[Figure]

Figure A5: Terrestrial precipitation recycling ratios according to UTrack using ERA5 input and NorESM2 input. A) Terrestrial moisture recycling ratio (%) from Tuinenburg et al. (2020), based on ERA5 data from 2008-2017. B) Terrestrial moisture recycling ratio (%) in the SSP2-4.5 scenario for 2015-2024.

[Figure]

Figure A6: Basin precipitation recycling ratios for the 26 major river basins of the world (also see Table 2) according to UTrack using ERA5 input and NorESM2 input. A linear regression (red solid line) gives $R^2$ = 0.62. The y=x line gives $R^2$ = 0.38. On average, the ERA5-based basin recycling ratios are 9.4 percentage point larger than those based on NorESM2.

Regarding the different set-up of the ERA5-based model: we appreciate this suggestion and agree it would be very interesting, but it lies outside the scope of this paper. This model transformation is not easy to do, the runs would still be very data-heavy and computationally expensive, and it would deserve a study on its own as we would have to do a systematic and global sensitivity analysis of all model changes. In Tuinenburg & Staal (2020, https://doi.org/10.5194/hess-24-2419-2020), we did perform some sensitivity analyses on the ERA5-based model version that we may use to interpret possible differences between the ERA5- and NorESM2-based runs.

This point is also interesting in relation to the sensitivity of the results to different ensemble members of NorESM2 (also see our reply to Ref. 1). We believe that extra sensitivity analyses and model comparisons are very important. Indeed, currently a moisture tracking

model intercomparison study is being carried out by an international community of moisture recycling modelers. This effort which will shed light on the importance of these model assumptions for moisture recycling results.

In addition, daily data from NorESM is used to force the UTrack model (Line 118: has a temporal resolution of one day). I assume that for wind fields and specific humidity instantaneous data is used and I wonder at which timestep of the day this data is taken? This is not stated in the methods and influences your results. If instantaneous wind fields are taken at midnight, features like a low level jet will enhance moisture transport, compared to instantaneous wind fields taken at noon. Opposite, sea-breeze features which enhance ocean-to-land moisture transport are mostly present during the day and thus will also influence your results when only one timestep during the day is used. This issue can be addressed by running UTrack with hourly ERA5 forcing, and with daily ERA5 forcing (as suggested in the previous point).

We agree that the daily resolution of the data from NorESM2 is a limitation. Diurnal fluctuations in winds, evapotranspiration and precipitation are thus averaged out. In case of evapotranspiration and precipitation, these are daily sums. However, indeed, the wind fields are instantaneous. We agree that this needs to be properly acknowledged. Therefore, we added the following text in lines 230-244 accompanied by the new supplementary figure A8:

"For NorESM2, we have one daily value for the wind field, an instantaneous value at 00Z. This may be a biased value compared to a higher temporal resolution of the daily cycle in wind speed. Wind speeds may be systematically different during different times of the day, which may lead to this bias. Moreover, because 00Z is at different solar (local) times around the globe, these biases may be spatially differing. Therefore, we estimated the bias in wind speed based on the ERA5 atmospheric reanalysis at different times of the day for the period 2010–2023. We retrieved the monthly mean reanalysis by time of day for the variables U and V between 1000 and 500 hPa. We calculated a quasi mean absolute wind speed between 1000-500 hPa based on these monthly values. Note that this is a 'quasi' wind speed, as we use the monthly mean U and V values, which is not the same as the monthly mean absolute wind speed. For both U and V, the hourly values have positive and negative values within a month, which will cancel out and thus not contribute to the absolute wind speed. In Fig. A8 we represent the absolute and relative difference of the 00Z quasi wind speed with the daily mean quasi wind speed. Typically, the absolute wind speed at 00Z deviates less than 0.2 m/s from the daily mean, although there are some regions with deviations up to 1 m/s. In relative terms, this deviation is typically within 5% of the wind speed, but with 10% deviation in some areas. A positive deviation will probably mean that the moisture recycling is underestimated, while a negative deviation will mean that the moisture recycling is overestimated. It is hard to translate these wind deviations to quantitative values of moisture recycling deviations, but given the low relative wind deviations, we expect the moisture recycling uncertainty due to this effect to be relatively small."

[Figure]

Figure A8: Deviations of instantaneous wind speed at 00Z with average hourly wind speed across the day in ERA5 based on the period 2010–2023. a) Relative deviations (%), b) Absolute deviations (m/s).

Last, I am a bit confused why the authors use a forced climate scenario to analyse past climate. I can follow the logic to take the scenario that is following the trajectory that the world is currently on (Fricko et al., 2017) (line 165), though citing a paper from 7 years ago feels a bit odd then. Wouldn't it be much more logic to use climate data forced with observed CO2 levels and observed SSTs?

We decided to do this for internal consistency, allowing us to isolate the relative differences in precipitation recycling during this century. We agree that citing a paper from less than 7 years ago is insufficient. Therefore, in addition to Fricko et al. (2017) explaining SSP2-4.5 to be a "middle-of-the-road" scenario, we added reference in line 187 to the Summary for Policy Makers in the latest IPCC Synthesis Report (2023), which explains how current climate policies would lead to a climate change trajectory consistent with SSP2-4.5.

**Introduction does not include all relevant literature and hypothesis**

In the introduction the authors state that (line 50): "However, where, how, and to which extent terrestrial moisture recycling will change in the future remains unclear." Although I agree there is still research to be done on how moisture recycling is changing to the future, there is also literature available already that addresses and (partly) answers this statement, and this literature is only cited in the discussion. An introduction needs to state the relevant literature on the topic and this is currently not done. Examples of literature that addresses the changes in moisture sources or moisture recycling in a future climate (Benedict et al., 2020; Findell et al., 2019; Fernandez-Alvarez et al., 2023). Furthermore, the introduction is also the moment to state hypothesis based on current literature, for example addressing the impacts of land-use change versus climate change. Currently the introduction provides more insights on the methods, describing the SSP scenarios and different moisture tracking methods which I found more relevant for the method section, or can be reduced.

Thank you. We used these useful references to add the following text in lines 46-48:

"It is expected that, on average, global warming will decrease terrestrial precipitation recycling ratios (Findell et al., 2019). However, regional differences in changes in land and sea sources of precipitation are likely (Fernández-Alvarez et al., 2023), as well as differences among seasons (Benedict et al., 2019; Fernández-Alvarez et al., 2023)."

**Improve results and discussion section**

In the first section of the results (line 187-192) the absolute and relative changes in land precipitation and land evaporation are given from NorESM. It would be very good to put these numbers in the perspective of a multi-model mean, for example given in Table 8.1 of Chapter 8 of the IPCC 2021 report, Douville et al., (2021). This indicates if the precipitation and evaporation averages from NorESM2 fall within or outside of the range of the CMIP6 multi-model mean.

This is a good suggestion. We added the following in lines 251-253: " … amounting to a 7% increase globally. This projection is typical for IPCC models, among which the average projected global precipitation increase by the end of the century is 6.6% (ranging between 3.3–11%) (IPCC, 2021).".

The result section reads as a bookkeeping exercise, where multiple alinea's (four alinea's from line 210 to line 245) have exactly the same structure but different numbers inserted for the different scenario's. This makes the result section dry and hard to read. This information suits well for a table instead, while in the text rather the interesting findings of the table can be reported.

Thank you. Also reviewer 1 suggested to include a table to summarize the results presented in (former) lines 211-245. We added this table as Table A4 and restructured this section to make it read less like a bookkeeping exercise.

Further, my suggestion would be to combine the results and discussion section to allow for direct comparison with literature. Currently, the result section is very dry as it is a sum-up of numbers and scenario's. By directly comparing moisture recycling ratios with the literature (combining results and discussion) allows for more perspective. At the moment, in the discussion the numbers from the result section are not repeated, which makes it very hard to put literature results next to the results of this study, which I think is very important to do. The same holds for the results on the major river basins, which could include more references to current literature.

We need to comply to journal requirements regarding paper structure, but we reorganize and rewrote the Results section to improve its readability.

On the discussion on the impact of land-use change and climate change on recycling ratio. I think this is a very interesting discussion point which is now addressed only shortly, I would dedicate a whole section on this. Are their different ways forward to test this influence? Some arguments that are provided later in the text (line 411) could potentially also be used to study impact land-use change vs climate change.

This is a good suggestion. We elaborated on this in new text in the Discussion (lines 504-512):

"Future work should use different methods to assess the relative contributions of climate change and land cover changes to global moisture recycling. In theory, evaporation over ocean should increase more rapidly than terrestrial evaporation due to their terrestrial soil moisture limitations (Findell et al., 2019). The increase in moisture input from the ocean will follow the Clausius-Clapeyron amplification (Fernandez-Alvarez et al. 2023). From the terrestrial part, soil moisture climatologies will generally show trends with future climate change (Lai et al., 2023), meaning that the absolute and relative contributions from ocean to precipitation should increase (for drying trends) or decrease (for wetting trends) with global

warming. Deviations from a soil-moisture based null model may point at the role of regional terrestrial conditions such as land cover changes. Also, moisture tracking forced by stylized climate- and land cover change experiments in Earth System Models would be a significant step forward in this regard."

**Substantial comments**

Line 40-50: Besides including relevant literature that assessed moisture recycling under a warming climate it is also good to address the impact of circulation changes on moisture recycling, such as changes in location of the ITCZ, Hadley cells, storm tracks, as this will affect the moisture transport as well.

We added reference to circulation changes in lines 45-46: "Global climate change will cause warming of the atmosphere as well as changes in atmospheric circulations (IPCC, 2021)." We also added in lines 148-149 the following statement about NorESM2: "The model also performs relatively well in reproducing multi-annual climatic variability such as the El Niño Southern Oscillation including El Niño teleconnections (Seland et al., 2020)."

Section 2.3 Simulations settings; I already addressed the issue of daily data in the major comments, but the limitation of only having limited pressure levels as input is not discussed at all. What is the impact of this on your results?

We believe the number of pressure layers (eight) is quite large, although it is smaller than in ERA5. A systematic assessment of this effect is beyond the scope of this study, but Tuinenburg & Staal 2020 (https://doi.org/10.5194/hess-24-2419-2020) did do some tests. Based on this we added the following sentence in lines 131-133: "Sensitivity tests done by Tuinenburg & Staal (2020) indicate that degrading the vertical moisture profile (here from 25 to eight) can affect moisture transport distances in the order of hundreds of km, but the number of pressure levels used here is still relatively large."

Line 176-184: Can the significant increases and decreases be quantified? Did you use a threshold to call it a significant increase or decrease?

Yes, we used $\alpha=0.05$. We added this missing information in line 201.

Section 2.4: When is the model initialized? As already mentioned I am a bit surprised that for current climate a scenario is used, while we already have the observations of current climate as the boundary conditions of the model. I could imagine this is done for consistency, but it would be nice to check how well the baseline run represent the actual conditions. Further, are these atmosphere only runs? So SST is prescribed?

Indeed, we do this for consistency. As explained above, we now use the results from Tuinenburg et al. (2020) to compare the global patterns of precipitation recycling against. The NorESM2 runs were coupled, without prescriptions of SST (Seland et al. 2020, https://doi.org/10.5194/gmd-13-6165-2020).

Line 174: Do I understand correctly that also for the SSP3-7 and SSP1-2.6 and SSP 3-7.0 you use the same climate sensitivity as SSP5-8.5? The approach here is unclear but you would expect that the climate sensitivity is used per SSP scenario to calculate changes in moisture recycling per degree warming.

We based it on only SSP5-8.5, because it has the largest increase in temperature, allowing for greater accuracy. We rephrased lines 195-197 for clarification: "For this we determined the global near-surface temperature rise for global land in NorESM2 , between the baseline and 2090–2099 in the SSP5-8.5 scenario, which was 4.7 °C."

Further, we are currently already warmer than the 1976-2005 baseline that is mentioned for which the 3.26 degrees is determined for. In the results a 3.26 change in temperature is used to move from the SSP2-4.5 (baseline; 2015-2024) to the SSP5-8.5, but I assume there is already some warming in the SSP2-4.5 baseline run for 2015-2024, which is now not taken into account. Thus if I understand the taken approach well, using the 3.26 degrees warming is incorrect.

This is correct and it may lead to a slight error. We improved this calculation by using the actual near-surface temperature data in NorESM2 (variable 'tas') for the 2015-2024 baseline in SSP2-4.5 and the 2090-2099 period in SSP5-8.5. This led to a larger estimated global temperature rise of 4.7 °C and this a lower estimated change in precipitation recycling ratio for each degree of warming, namely 1.5% per degree of warming. We updated the numbers throughout the manuscript. Thank you for this suggestion.

Results

Line 177: "wetting, land dominated" if a significant increase in precipitation coincides with a significant **decrease** in terrestrial precipitation recycling --> should this not be exactly opposite? An increase in terrestrial precipitation recycling? Line 461-462 also states that land-dominated means increase in terrestrial precipitation recycling

Thank you for spotting these typos, they indeed had to be the opposite.

I would suggest to leave the min and max value of moisture recycling out of the text to make it more readable. Those min and max values could be reported in a table. By providing the std you give an idea of the interannual variability in the text.

We removed the minima and maxima from the main text and included them in Table 1, as well as the standard deviations. We removed the results for the 2050s from this table, because they were not discussed in the main text.

Line 213-215; How relevant is it to give this information if it only concerns such a small percentage of land grid cells (1.1% and 1.3%)? Instead, it would be nice if some words are dedicated on where those 8.7% of land grid cells are on the globe that show a significant change in precipitation. And do you mean with precipitation absolute precipitation or precipitation recycling? In line 170 it is stated that statistical significance is tested for precipitation recycling, but from the result section it reads as if it is checked for absolute precipitation changes, which is confusing. I read in the caption of Figure 2 and 3 it is about significant differences in precipitation recycling, it would be nice to have those regions that show a significant change are hatched in figure 2 and 3.

We believe that even if it represents a quite small area of the global land, these numbers are still relevant to report. We agree, though, that some more detail about where the areas with significant changes are located is good to add, so we added in lines 305-306:

"Drying is mostly concentrated in the Amazon and eastern Europe; wetting occurs mostly in the high northern latitudes and in eastern Asia."

Line 279-277: I am not sure about the purpose of this alinea. Are these findings of this study? Or are these numbers given here to indicate the impact of land-use change on moisture recycling? If so, I would discuss them in combination with the discussion section 4.1.

These are numbers we calculated, but in essence are not novel results. Indeed we provide these numbers so we can better interpret the land-cover versus climate change effects on moisture recycling. We shortened this to only provide the most essential information and integrated it into the first paragraph of the Results, where the overall global patterns in precipitation and evaporation (and now including forest and cropland cover) are provided.

**Minor comments / typos / small unclarities:**

Define precipitation recycling in abstract

We rephrased the first sentence of the Abstract as follows: "Many areas across the globe rely on upwind land areas for their precipitation supply through on terrestrial precipitation recycling, which is the amount of precipitation that has evaporated from upwind land areas."

Line 20: moisture recycling ratio--> do you mean with moisture recycling precipitation recycling? Terms are used throughout it each other but it is unclear what is what

Thank you, we changed to "global terrestrial precipitation recycling".

Line 108: For equations  --> equations of what? Of the moisture tracking model?

Yes. We specified this in line 116.

Line 134: 'There is some overestimation of global mean temperature' --> this is very vaguely stated, can you quantify?

We removed this sentence, as the main point was given in the previous sentence.

Line 148: 'We used these forcing data directly without interpolation' --> How can you have daily data and run the model on 4-hourly timesteps, without interpolation?

We added the following sentence in lines 162-164: "Individual moisture parcels may cross multiple grid cells during one time step if the time step is too large. This may cause errors in the parcel trajectories, which is solved by taking a sufficiently small time step, even if the data themselves are not interpolated."

Line 159: In Line 153 it says 1000 parcels per mm, and here 100 parcels per mm

Correct. We used different settings for the global runs and the basin runs, based on the fact that the areas of the two differ by orders of magnitude. Effectively, the 100 parcels per mm in the basin runs mean that the same parcel already represents a larger volume of water in the basin runs compared to the 1000 parcels in the global run. We added the following sentences in lines 180-182 to clarify this: "Instead of 1000 parcels for every mm of evaporation globally, we released 100 parcels for every mm of evaporation from the each basin. Note that this implies a larger amount of parcels per volume than in the global runs due to the considerably smaller source area of the basins."

3.1 Global land --> can you make the headers more self-explanatory? The result section will also benefit from more section and section headings to illustrate the red-threat

We made the headings more self-explanatory, but retained the division between global results and results for the major river basins, as further separation of section 3.1 would result in sections of single (small) paragraphs.

Line 492: less clouds but more rain? Maybe I misunderstand

We deleted "likely linked to underestimated cloud cover", as the point simply is that precipitation over oceans is overestimated in NorESM2.

Line 499: 'Around one-fifth of global precipitation is attributed to vegetation' --> to me it is not clear what is meant with this sentence

We rephrased (lines 506-507) to: "Around one-fifth of global precipitation has been evaporated directly by vegetation (Keys et al., 2016)."

Conclusions

Line 538-539: 'widespread drying accompanied by disproportional reductions of moisture supply over land' à this sentence counteracts the argument of global greening and increased evaporation (stated in the discussion)

We meant regional-scale drying, such as in the Amazon and eastern Europe, which contrasts with the global average. To make this more clear, we replaced "widespread" by "regional".

Here the word disproportional is often used, what is meant with that?

We meant that the relative change in moisture supply exceeds the relative change in precipitation, but we deleted both instances of the word in the Conclusions, as the message of the respective sentences did not depend on them.

Line 541-542: can you back-up this last sentence by findings from the study?

This was a high-level take-away based on the discussion of our results in the context of the literature. However, we now stick more closely to the results and instead highlight the differences among the SSPs, as follows: "Although both land cover changes and global climate change affect moisture recycling over land areas, most of the differences in moisture recycling among the various SSPs over the 21$^{st}$ century seem to be caused by direct and indirect effects of global warming."

Figures

Figure 2 to 6: These figures can be improved and made more readable by only displaying one legend (colorbar) per figure, and not for all subfigures. In this way the figures can be enlarged.

Good suggestion, we did that with all figures including the supplementary ones (where relevant).

---

## Author Response (AR2)

Dear Editor,

Thank you for your careful review. Below we reply to your final comments. In addition to these revisions, we fixed an error in the Abstract, where "2.1%" was not yet replaced by "1.5%" in the previous revision.

We hope the manuscript is now ready for publication and we look forward to your positive assessment.

On behalf of all authors,

Kind regards,

Arie Staal

* Line 227, " implying that long-term climate variability in NorESM2 does not affect our main outcomes." should probably be "implying that internal variability in NorESM2 does not affect our main outcomes at the global scale."

Thank you for spotting this, we adjusted as suggested.

* Thinking along these lines, Figure A7 is a good addition to check whether 10 year periods are sufficient to robustly sample forced climatic changes, rather than random changes due to internal variability. However, you performed this analysis at the global scale, where averaging has substantially improved signal-to-noise. Please assess if the grid point-based analyses and river basin-based analyses are equally robust, for all SSPs.

This is a good point, so we analyzed the terrestrial precipitation recycling ratios of the 26 major river basins in the same way as we did for the global land area. We find that the robust patterns between 10- and 30-year time slices that we found for the global land are also present in all of these river basins. Below, in Figures R1-R4, we paste a few examples of major river basins at different continents: the Amazon, Congo, Mississippi and Yangtze basins. We decided, however, not to include these additional figures in the supplement of the paper itself, to limit the figures to the most essential ones. Thank you in advance for your understanding.

[Figure]

Figure R1: Moving averages of the global terrestrial precipitation recycling ratio (TPR, in %) ± one standard deviation for ten-year time slices (in red) and 30-year time slices (in blue) in the Amazon basin, for (a) SSP1-2.6, (b) SSP2-4.5, (c) SSP3-7.0, and (d) SSP5-8.5.

[Figure]

Figure R2: Moving averages of the global terrestrial precipitation recycling ratio (TPR, in %) ± one standard deviation for ten-year time slices (in red) and 30-year time slices (in blue) in the Congo basin, for (a) SSP1-2.6, (b) SSP2-4.5, (c) SSP3-7.0, and (d) SSP5-8.5.

**Mississippi**

[Figure]

Figure R3: Moving averages of the global terrestrial precipitation recycling ratio (TPR, in %) ± one standard deviation for ten-year time slices (in red) and 30-year time slices (in blue) in the Mississippi basin, for (a) SSP1-2.6, (b) SSP2-4.5, (c) SSP3-7.0, and (d) SSP5-8.5.

**Yangtze**

[Figure]

Figure R4: Moving averages of the global terrestrial precipitation recycling ratio (TPR, in %) ± one standard deviation for ten-year time slices (in red) and 30-year time slices (in blue) in the Yangtze basin, for (a) SSP1-2.6, (b) SSP2-4.5, (c) SSP3-7.0, and (d) SSP5-8.5.

* Figure 2- units along side the colour bar should be % or %-point I think.

Thank you for spotting this. We fixed this error (without applying tracked changes).

* In the discussion you state, line 631: "Three ensemble members of historical runs gave around 0.1 °C difference in global temperature on a yearly basis (Seland et al., 2020)." I don't believe this is true, ENSO variations in these runs should lead to much larger difference at yearly timescales.

We removed this sentence so the text that was added in the previous revision now reads: "Furthermore, the model generates internal climate variability. Ensemble runs allow for the separation of internal climate variability from the forced climatic signal. In the future, different ensemble members can be used to force UTrack, so the signal and noise in precipitation recycling can be separated, even though it is unlikely that the global trends and patterns would be affected by different ensemble members."